# Spatial proteomics reveals secretory pathway disturbances caused by neuropathy-associated TECPR2

Karsten Nalbach[1], Martina Schifferer [2,3], Debjani Bhattacharya[1], Hung Ho-Xuan [4], Wei Chou Tseng[5], Luis A. Williams[5], Alexandra Stolz [4], Stefan F. Lichtenthaler [2,3,6], Zvulun Elazar [7] & Christian Behrends [1]✉

Hereditary sensory and autonomic neuropathy 9 (HSAN9) is a rare fatal neurological disease caused by mis- and nonsense mutations in the gene encoding for Tectonin β-propeller repeat containing protein 2 (TECPR2). While TECPR2 is required for lysosomal consumption of autophagosomes and ER-to-Golgi transport, it remains elusive how exactly TECPR2 is involved in autophagy and secretion and what downstream sequels arise from defective TECPR2 due to its involvement in these processes. To address these questions, we determine molecular consequences of TECPR2 deficiency along the secretory pathway. By employing spatial proteomics, we describe pronounced changes with numerous proteins important for neuronal function being affected in their intracellular transport. Moreover, we provide evidence that TECPR2's interaction with the early secretory pathway is not restricted to COPII carriers. Collectively, our systematic profiling of a HSAN9 cell model points to specific trafficking and sorting defects which might precede autophagy dysfunction upon TECPR2 deficiency.

Hereditary sensory and autonomic neuropathies (HSAN) are a diverse group of rare neurodevelopmental and degenerative disorders sharing progressive loss of autonomic and sensory peripheral nervous system function. The most recently identified subtype, HSAN9, is caused by mutations in the gene encoding Tectonin β-propeller repeat containing protein 2 (TECPR2). This autosomal recessive disease is clinically characterized by global developmental delay, intellectual disability as well as severe autonomic and sensory disfunction. Due to additional progressive spasticity and paralysis of the lower limbs, the disease was initially classified as spastic paraplegia 49 (SPG49)[1–3]. Affected individuals suffer from recurrent respiratory infections and central apnea which often lead to death by the age of 20[1–4]. TECPR2 consists of an N-terminal tryptophan-aspartic-acid (WD) 40 domain, a series of

enigmatic TECPRs and a LC3-interacting region (LIR)[5]. Several pathogenic nonsense and missense mutations have been identified in TECPR2[4] such as c.1319delT which leads to a frameshift and premature stop. To what extend the resulting truncated TECPR2 variant (L440Rfs*19) that lacks more than half of the protein including the TECPR and LIR motifs is endogenously expressed and stable in cells remains unclear since antibodies specifically recognizing the N-terminal part of TECPR2 are not available. As we currently lack this type of information for any of the TECPR2 mutations, the contribution of loss or gain of function mechanisms to HSAN9 pathogenesis is largely unknown.

TECPR2 was identified as an interactor of human autophagy-related 8 (hATG8) proteins[5] and a modulator of autophagy[1,5]. However,

[1]Munich Cluster for Systems Neurology (SyNergy), Medical Faculty, Ludwig-Maximilians-University München, Munich, Germany. [2]German Center for Neurodegenerative Diseases (DZNE), Munich, Germany. [3]Munich Cluster for Systems Neurology (SyNergy), Munich, Germany. [4]Buchmann Institute for Molecular Life Sciences and Institute of Biochemistry II, Faculty of Medicine, Goethe University Frankfurt, Frankfurt, Germany. [5]Q-State Biosciences, 179 Sidney Street, Cambridge, MA 02139, USA. [6]Neuroproteomics, School of Medicine, Klinikum rechts der Isar, Technical University of Munich, Munich, Germany. [7]Departments of Biomolecular Sciences, The Weizmann Institute of Science, Rehovot, Israel. ✉e-mail: christian.behrends@mail03.med.uni-muenchen.de

the precise step of TECPR2's involvement in this pathway is poorly understood. The impairment of autophagic flux with accumulated autophagosomes and reduced numbers of autolysosomes[1,5,6] as well as the interaction with the homotypic fusion and protein sorting (HOPS) complex and the biogenesis of lysosome-related organelles complex 1 (BLOC-1) indicate a possible role of TECPR2 in autophagosome-lysosome targeting or fusion[5,6]. However, TECPR2 was also shown to bind and stabilize SEC24D[5], a core component of COPII carriers which facilitate protein export from the endoplasmic reticulum (ER) at ER exit sites (ERES) and transport cargo towards the Golgi[7]. COPII carriers are composed of an inner (SAR1, SEC23, SEC24) and outer (SEC13, SEC31) coat[8,9] and assembly as well as transport is regulated by multiple factors including SEC16A, MIA3, TFG and the transport protein particle (TRAPP) tethering complex[10]. As expected from its interaction with SEC24D, TECPR2 is required to maintain ERES and regulates ER export of several exogenously expressed trafficking reporters[5]. Consistent with emerging evidence that ERES and COPII carriers contribute to autophagosome formation[11-14], TECPR2 also affected the formation of early pre-autophagosome structures[5]. Given that proper lysosome function requires a constant influx of its components via different trafficking routes which originates at the Golgi[15], it is possible that TECPR2's role in autophagy might be consequential of its involvement in ER-to-Golgi transport.

To start testing this hypothesis, we sought to determine which proteins depend on TECPR2 for their trafficking out of the ER and sorting within the cell. Importantly, an inventory of cargo proteins that accumulate off pathway when TECPR2 is defective will help to identify cellular processes in addition to autophagy that might contribute to the pathogenesis of HSAN9. Hence, we employed a series of spatial proteomic approaches to systematically examine defects along the secretory pathway in an HSAN9-mimicking cell model. This global analysis unveiled profound alterations at distinct trafficking steps including ERES formation, COPII-mediated transport to the Golgi, sorting to the lysosome or PM and secretion into the extracellular space. Intriguingly, a number of proteins affected at these compartments are potentially relevant for HSAN9. In addition, we identified a number of ER- and Golgi-associated binding partners of TECPR2, most of which are lost or dramatically reduced when TECPR2's C-terminus is missing, thus supporting a loss-of-function mechanism for HSAN9. Overall, we provide a comprehensive resource which will facilitate the mechanistic dissection of TECPR2 and enable biomarker development for studying HSAN9 and related diseases which involve secretory pathway defects.

## Results

### HSAN9-truncated TECPR2 alters the assembly and local environment of ERES

To dissect the dependency of the early secretory pathway on TECPR2, we used CRISPR/Cas9 technology to generate 293T cells carrying a TECPR2 frame shift mutation that mimicked the HSAN9-associated L440Rfs*19 variant. Similar to HSAN9 patient-derived fibroblasts[6], full-length TECPR2 was undetectable in these cells which were referred to as TECPR2 mutant (MUT) (Fig. 1a). Using this model system, we took an APEX2-based proximity biotinylation approach to explore the impact of disease-associated TECPR2 on the composition of ERES (Fig. 1b). Thereto, a panel of seven core ERES components (SEC12, SEC16A, SEC23A, SEC23B, SEC24A, SEC13, SEC31A) was individually fused to myc-tagged APEX2 and expressed in TECPR2 WT and MUT cells (Supplementary Fig. 1a) where they partially colocalized with endogenous ERES components (Supplementary Fig. 1b). Upon induction of biotinylation, cells were lysed and biotinylated proteins enriched by streptavidin followed by on-beads tryptic digestion and MS analysis (Fig. 1c). Across the different ERES baits, quadruplicate samples showed high correlation ($r \geq 0.89$) (Supplementary Fig. 2a). In total, we identified 3536 proteins of which 1070 proteins passed filtering for

contaminants and were enriched above unspecific background (Supplementary Fig. 2b). Gene ontology (GO) annotation analysis of these proteins unveiled numerous secretory pathway-associated terms, confirming the specificity of our approach (Supplementary Fig. 2c). Depending on the APEX2 chimera, between 309 and 752 specific proteins were identified (Supplementary Fig. 2d), among which ERES components were detected as common proximity partners in almost all conditions (Supplementary Fig. 2e). Comparative analysis of proteins detected in the proximity of APEX2-tagged ERES components revealed significant changes between TECPR2 WT and MUT cells (Fig. 1d, Supplementary Data 1). Intriguingly, for almost all baits the spectrum of proximity partners was considerably larger when TECPR2's C-terminus was absent (red proteins) (Fig. 1d), possibly reflecting an increase in non-specific associations due to disintegration of ERES. Consistent with this notion, GO analysis of similarly altered proteins showed that several terms such as ER membrane and cytoplasm were differently enriched in TECPR2 WT and MUT cells, respectively (Fig. 1e). A closer look at known ER-Golgi transport proteins unveiled an enrichment of USO1, GOLGA2, STX5 and RAB1B in the neighborhood of a number of different ERES baits in TECPR2 MUT cells while factors such as MIA3, SEC16A, SEC24A, SEC24B, SEC24C, SEC22B, SEC23B, SEC31A, BET1, PEF1 and PDCD6 were all decreased in this condition (Fig. 1f). In a parallel whole cell proteomics experiment, we globally quantified protein abundance changes and found that the vast majority of proteins - including ER-Golgi transport factors (Fig. 1f) - remained largely unaltered upon expression of truncated TECPR2 (Supplementary Fig. 3a–f, Supplementary Data 2). As reported previously by RNAi experiments[5], immunofluorescence analysis unveiled significantly decreased numbers of SEC24C- and SEC13-positive puncta in TECPR2 MUT cells which could be rescued by reintroducing HA-tagged TECPR2 WT but not by the disease-mutant L440Rfs (Fig. 2a, b). Notably, re-expression of these TECPR2 variants did not affect the morphology of the ER or the Golgi (Supplementary Fig. 3g). The robust reduction of SEC24C- and SEC13-positive puncta was also observed in HSAN9 patient-derived fibroblasts (Fig. 2c, d). Consistent with the notion of disintegrated ERES, colocalization of different COPII subunits was severely diminished in cells lacking full-length TECPR2 (Fig. 2e). Conversely, SEC13 was found biochemically in closer proximity to the cis-Golgi components GOLGA2, STX5 and USO1 upon TECPR2 deficiency (Fig. 2f). Besides, our profiling identified a considerable number of proteins that has previously not been linked to ERES or ER-Golgi transport but showed significant proximity changes in TECPR2 MUT cells across almost all ERES baits (Fig. 1f, Supplementary Data 1). For the neurological disorder-associated proteins SPG20 and NEK9 as well as the co-chaperone BAG2, these alterations were confirmed by immunoblotting for biotinylated proximity partners of APEX2-SEC13 and -SEC16A in TECPR2 WT and MUT cells (Fig. 2g, h). Notably, SPG20, NEK9 and BAG2 lack clear signal peptides and were found to be protease-sensitive in protection assays (Fig. 2i), suggesting that these proximity partners are unlikely secretory cargo proteins which accumulate in response to compromised ERES. Together, these results indicate that the assembly and/or maintenance of ERES requires full-length TECPR2 and is disturbed by HSAN9-associated TECPR2.

### HSAN9-truncated TECPR2 affects ER exit and trafficking of COPII cargo

As the majority of proteins exits the ER through COPII-coated carriers en-route to the Golgi, we next set out to identify COPII-dependent cargo whose trafficking is impaired in cells mimicking HSAN9-truncated TECPR2. For this purpose, we combined the retention using selective hook (RUSH) system[16] with protease protection-enhanced APEX2-mediated proximity biotinylation[17] to map the content of COPII carriers (Fig. 3a). Briefly, APEX2 was fused to the luminal

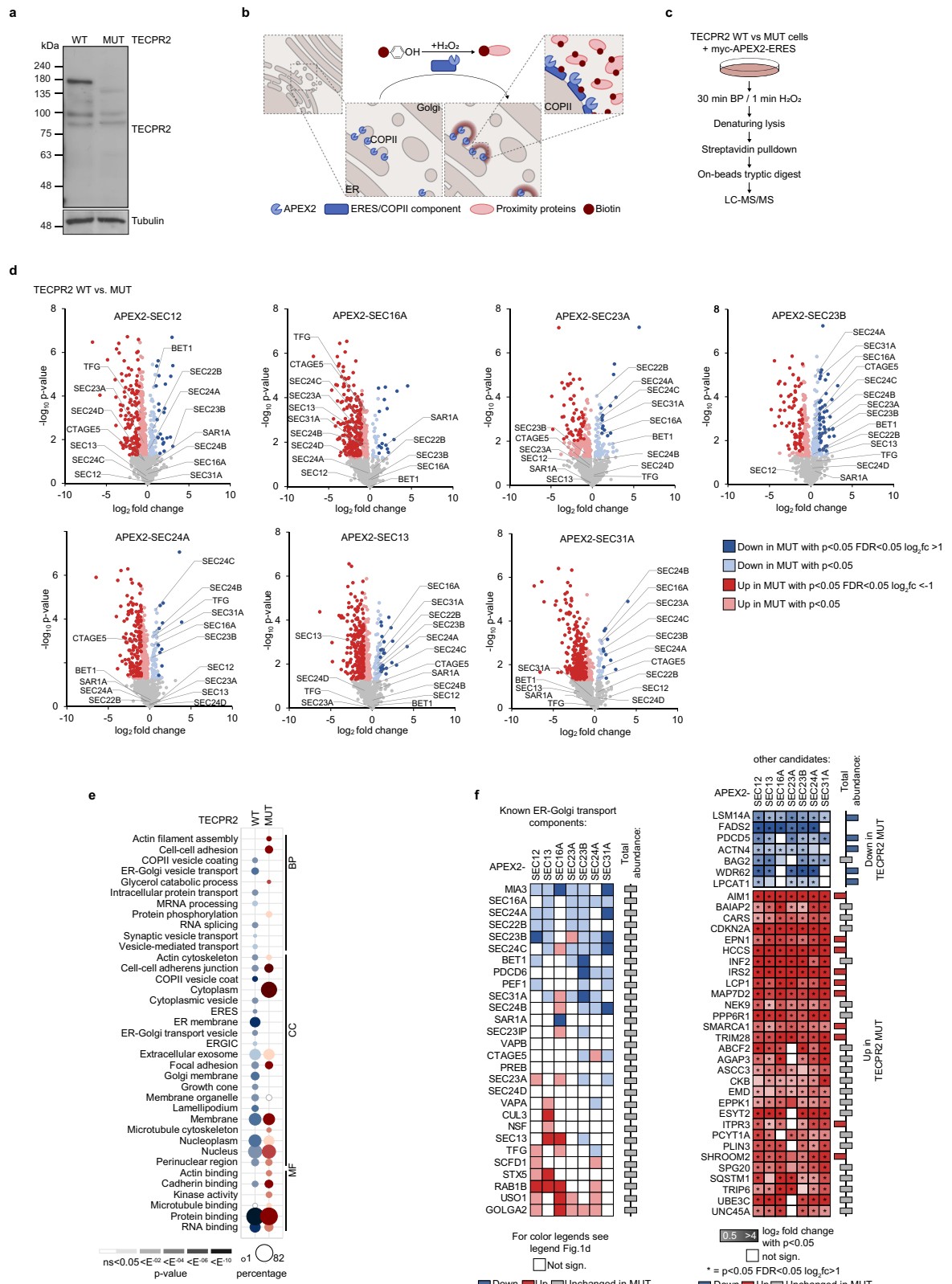

C-terminus of an α-mannosidase II-based RUSH reporter (MAN2A1-SBP-GFP) and introduced in TECPR2 WT and MUT cells. The reporter was released from its ER hook upon incubation with biotin-phenol (BP) at varying conditions, representing different trafficking states (30 min at 4 °C for ER localization, 5 min at 37 °C for COPII localization and 30 min at 37 °C for Golgi localization). Within the last minute of each condition, biotinylation was triggered by $H_2O_2$ pulsing. After

homogenization, cytosolic proteins were cleared by proteinase K, protective membranes were lysed in denaturing conditions and biotinylated proteins were enriched via streptavidin. Samples were subjected to tryptic digestion followed by MS (Fig. 3b). Prior to proteomic profiling, we validated expression (Supplementary Fig. 4a), efficient biotinylation (Supplementary Fig. 4b), proteinase K protection (Fig. 3c) and synchronized ER-Golgi trafficking (Fig. 3d) of RUSH-APEX2.

**Fig. 1 | Mutant TECPR2 alters the assembly and proximity of ERES components.**
**a** TECPR2 WT and MUT 293T cells were lysed and analyzed by immuno-blotting. **b** Proximity biotinylation of ERES. **c** Proteomics workflow. **d** Volcano plots of proteins detected by APEX2-ERES in TECPR2 WT and MUT cells. Significantly altered proteins are labeled in dark color as class I or light color as class II (two-sided t-test, $n = 4$ independent experiments). Known ER export factors are highlighted. **e** GO terms of class I proteins significantly enriched in the proximity of at least four and six different baits in TECPR2 WT and MUT cells, respectively. Dot size correlates to number of proteins, dot color to term enrichment (*p*-value). **f** Detection of known ER-Golgi transport factors by APEX2-ERES (left panel) and of other candidates

enriched in the proximity of at least four and six different APEX2-ERES components in TECPR2 WT and MUT cells (right panel). In the left panel, color code indicates class I (dark) or class II (light) proteins as in d. Empty boxes indicate no change. Right-handed bars represent change of depicted ERES components in whole cell proteomics comparing TECPR2 WT and MUT cells. Right panel, color code refers to log2 fold change of significantly altered proteins. Empty boxes indicate no change. * highlights class I proteins. Right-sided bars represent change of respective proteins in the whole cell proteomics analysis comparing TECPR2 WT and MUT cells (blue, decreased in MUT; red, increased in K; gray, not changed). Source data are provided as a Source Data file.

Quantitative analysis of the latter unveiled a delay in ER-Golgi trafficking of RUSH-APEX2 in cells harboring truncated TECPR2 (Fig. 3e) as previously reported for the RUSH 'only' reporter in TECPR2 depleted cells[5]. Using our RUSH-APEX2 approach, we identified 980 non-redundant proteins in four biological replicates per condition of which 500 passed filtering for contaminants and proteinase K resistant proteins with high reproducibility (Supplementary Fig. 4c, d, Supplementary Data 3). The vast majority of these protease-protected proteins were enriched compared to non-biotinylated controls across the different conditions, detected after initiation of trafficking at 37 °C (Supplementary Fig. 4e) and possessed secretion-related GO term annotations (Supplementary Fig. 4f). Analysis of the relative abundance of candidates in the different trafficking compartments revealed that known ER-associated proteins tended to be enriched after 5 min trafficking, while known Golgi-associated proteins were mostly enriched after 30 min (Supplementary Fig. 4g). Truncated TECPR2 caused a marked delay in this pattern (Supplementary Fig. 4g), mirroring the results from the image-based analysis (Fig. 3e). Examining the overlap between 5 min and 30 min trafficking at 37 °C showed that the majority of biotinylated proteins was already enriched after 5 min in TECPR2 WT cells whereas a substantial number of candidates were only identified after 30 min in TECPR2 MUT cells (Supplementary Fig. 4h). A large fraction of the latter candidates showed enrichment of the GO term annotations 'ER' or 'secreted' (Supplementary Fig. 4i), further strengthening a potential delay in ER export in HSAN9 mimicking conditions. Direct comparison of biotinylated proteins detected in TECPR2 WT and MUT cells at different trafficking steps showed that a majority of these proteins decreased in their local abundance in COPII carriers in TECPR2 MUT cells across all conditions, especially after 5 min and 30 min trafficking at 37 °C (Fig. 3f and Supplementary Fig. 4j). The overlap of these latter two conditions contained 108 common proteins (Fig. 3g) whose total cellular abundance largely remained unchanged (Fig. 3h, Supplementary Data 2) and which carried prominent ER-, Golgi-, extracellular-, glycosylation- and transmembrane-related GO term annotations (Fig. 3i). Moreover, several structural features shared by proteins involved in establishment of the extracellular matrix and cell-cell communication including the EGF domain, the N-terminal Laminin domain and the Plexin, Semaphorin and Integrin (PSI) domain were likewise enriched in this overlap (Fig. 3i). From the pool of COPII cargo candidates, we selected CLPTM1L, GOLIM4, ERGIC1 and LGALS3BP as secreted, multi-pass and single-pass transmembrane proteins, respectively, and monitored their subcellular distribution in non-synchronized cells at endogenous levels. While there was no consistent trend in total abundance changes of these candidates (Fig. 3j and Supplementary Fig. 4k), expression of truncated TECPR2 resulted in a more diffuse and less punctate appearance of CLPTM1L, ERGIC1 and LGALS3BP while distinct puncta emerged from condensed GOLIM4-positive structures (Fig. 3j). Taken together, these findings show that combining RUSH and APEX2 systems enables to map the content of COPII carriers and to systematically identify a broad range of COPII cargo candidates whose ER export and subsequent trafficking is impaired upon loss of full-length TECPR2.

## HSAN9 truncated TECPR2 alters the cell surface proteome and secretome

Complementarily to our analysis of ERES and COPII carriers, we next assessed to what extent lack of full-length TECPR2 affects the arrival of proteins at their final secretory pathway destination. As a starting point, we focused on the plasma membrane (PM) and the extracellular space. To examine changes in these two compartments, we employed click chemistry-mediated biotinylation of metabolically labeled glycoproteins. These were enriched from cell lysates for surface-spanning protein enrichment with click sugars (SUSPECS)[18] or from concentrated media following extraction with Concanavalin A for secretome protein enrichment with click sugars (SPECS)[19]. In both approaches, biotinylated proteins were digested with trypsin and identified by MS (Fig. 4a). As quality control, we validated the enrichment of biotinylated proteins from the PM and media as well as biotinylation of glycoproteins at the surface of TECPR2 WT and MUT cells upon labeling (Supplementary Fig. 5a–c). Cell surface proteome and secretome analysis of these cells unveiled 1330 and 313 proteins (Fig. 4b–d), respectively, that were enriched over non-biotinylated controls (Supplementary Fig. 5d, e, Supplementary Data 4 and Supplementary Data 5) with high reproducibility (Supplementary Fig. 5f, g) and specificity for glycoproteins, PM, secreted and signal peptide-containing proteins (Supplementary Fig. 5h, i). As an additional stringency filter, we also performed SUSPECS and SPECS with TECPR2 MUT cells reconstituted with HA-TECPR2 WT or L440Rfs (Fig. 4a). Strikingly, a substantial fraction of proteins whose abundance significantly changed in the PM or media in TECPR2 MUT cells showed the same significant alterations in HA-TECPR2 L440Rfs expressing cells (Fig. 4b, Supplementary Fig. 5j, k), providing further support that truncated TECPR2 fails to support its function in the secretory pathway. GO term analysis revealed a prominent reduction of neuronal annotations among cell surface proteome and secretome constituents in TECPR2 MUT cells, whereas endosome-, lysosome- and collagen-related terms were enriched (Fig. 4e, f). A comparison of annotation enrichment between the click-chemistry and COPII profiling approaches revealed alteration of similar structural terms including Laminin, Fibronectin, EGF-like and PSI domains upon TECPR2 deficiency (Figs. 3i and 4e, f), indicating that proteins carrying these features might be particularly sensitive to altered TECPR2 function. Importantly, a large portion of altered SUSPECS and SPECS candidates were unchanged in their total abundance (Fig. 4g, h, Supplementary Data 2). To validate altered candidates represented by these GO annotation categories, we employed differential centrifugation-based subcellular fractionation and monitored TECPR2-dependent PM abundance changes of GOLIM4, M6PR, NCAM1, PLXNA1 and PLXNA2 in TECPR2 MUT cells (Fig. 4i) and patient-derived fibroblasts (Supplementary Fig. 6a). While GOLIM4 and M6PR seemed to be missorted or retained at the PM in the absence of full-length TECPR2, PLXNA2 and NCAM1 were not found at this compartment in any substantial amounts. Notably, we confirmed the latter finding by confocal microscopy. NCAM1 was almost completely absent from the PM in TECPR2 MUT cells while the localization of ATP1A1 as a control protein was not affected (Fig. 4j). Similar results were obtained in TECPR2 deficient HeLa and SH-SYS5 cells

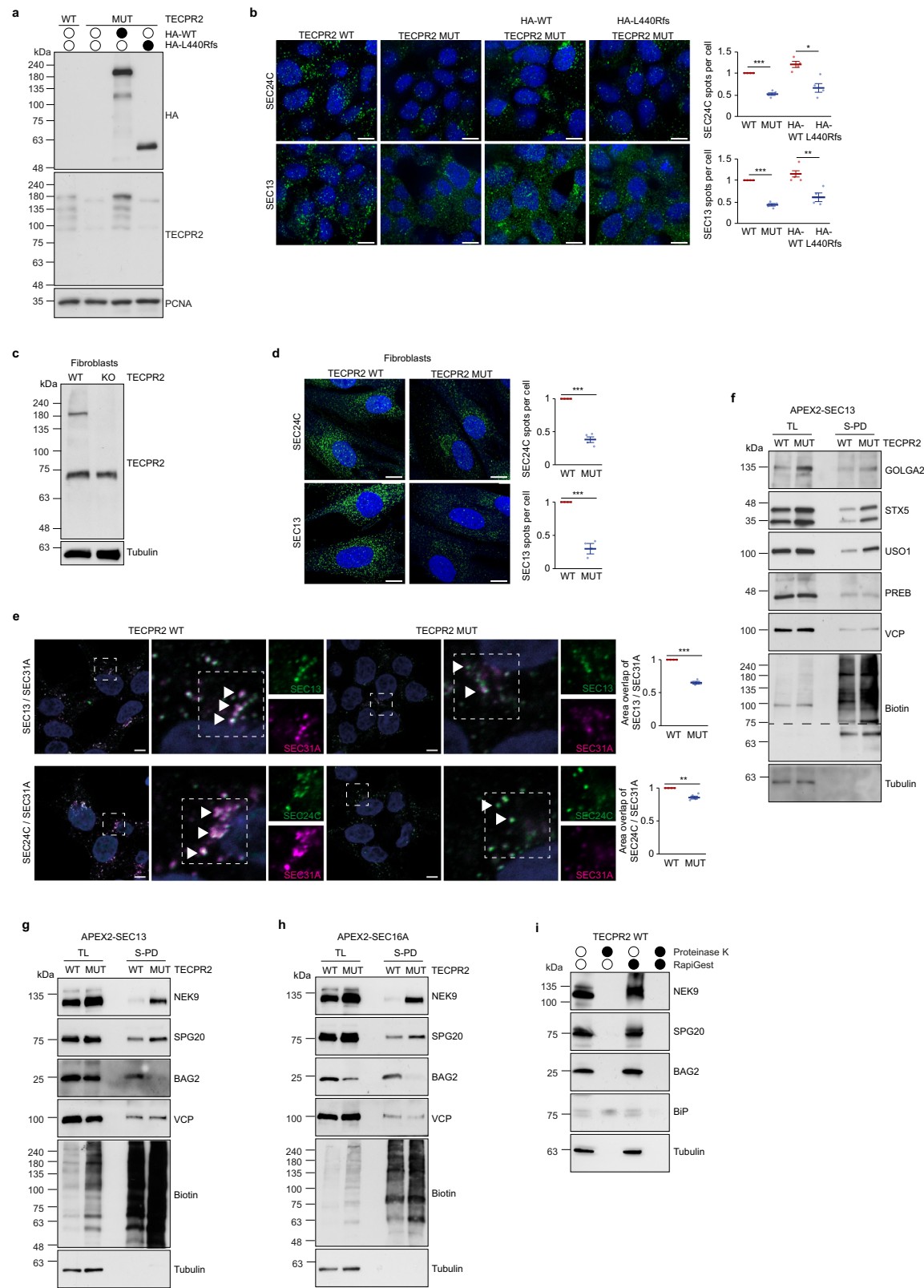

(Supplementary Fig. 6b–e). Lastly, analysis of media from TECPR2 MUT cells and patient-derived fibroblasts revealed an increase in secreted NID1 and HS6ST2 while the amount of CRLF1, EDIL3 and TNC were decreased in (Fig. 4k, Supplementary Fig. 6f). To account for the contribution of unconventional protein secretion (UPS) to the observed secretome changes, we compared our SPECS results with two published datasets on autophagy-dependent and exosome-

dependent UPS[20,21]. Although a small proportion of SPECS candidates were also present in these datasets, the vast majority of them was not identified to be secreted in an autophagy- or exosome-dependent manner (Supplementary Fig. 6g, h). Taken together, SUSPECS and SPECS proteomics demonstrate that likely due to disturbed early secretory processes, arrival of cargo at their final trafficking destination is also affected in our disease-mimicking TECPR2 cell model.

**Fig. 2 | ERES disintegrate in TEPCR2 deficient cells. a** Immunoblot analysis of TECPR2 WT and MUT cells as well as TECPR2 MUT cells re-expressing TECPR2 WT and L440Rfs. PCNA served as loading control. **b** TECPR2 WT and MUT cells were fixed and immunolabeled with anti-SEC13 or -SEC24C. Nuclei were stained with DRAQ5. Statistical two-sided t-test analysis of normalized SEC24C and SEC13 spots (*n* = 4 independent experiments). Error bars represent mean ± SEM, *p* value = 0.00003 and 0.01064 for SEC24C and *p* value <0.00001 and =0.00846 for SEC13. Scale bars 10 μm. **c** Immunoblotting of fibroblasts expressing WT or L440Rfs*19-mutant TECPR2. **d** Fibroblasts were fixed and immunolabeled with anti-SEC13 or -SEC24C. Nuclei were stained with DRAQ5. Statistical two-sided t-test analysis of normalized SEC24C and SEC13 spots (*n* = 4 independent experiments). Error bars represent mean ± SEM, *p* value = 0.00001 for SEC24C and *p* value = 0.00021 for SEC13. Scale bars 10 μm. **e** TECPR2 WT and MUT cells were fixed and immunolabeled with anti-SEC13 or -SEC24C together with an anti-SEC31A antibody. Insets show magnification of boxed areas. Scale bars 10 μm. Quantification of area overlap SEC13/SEC24C with SEC31A (normalized to WT, two-sided t-test analysis, *n* = 4 independent experiments, error bars represent mean ± SEM, *p* value <0.00001 for SEC13 and *p* value = 0.00699 for SEC24C.). **f** TECPR2 WT and MUT cells expressing APEX2-SEC13 were subjected to biotinylation followed by lysis and streptavidin pulldown (S-PD). TL, total lysates. TECPR2 WT and MUT cells expressing APEX2-SEC13 (**g**) or APEX2-SEC16A (**h**) were subjected to biotinylation followed by lysis and streptavidin pulldown (S-PD). TL total lysates. **i** Homogenates from TECPR2 WT cells were left untreated or incubated with proteinase K, RapiGest or both. BiP and Tubulin served as controls. Source data are provided as a Source Data file.

Intriguingly, several proteins associated with neuronal development and maintenance (e.g., NCAM1, TNC) were among the altered cargo at the PM or in the extracellular space. Hence, our findings may provide a potential link between defects in ER export and the neurodevelopmental and -degenerative phenotypes observed in HSAN9 patients.

## HSAN9-truncated TECPR2 impacts the composition of lysosomes

Another cellular destination that can be reached by trafficking through the early secretory pathway are lysosomes. Since a recent study reported changes in lysosomal morphology upon loss of TECPR2[6], we thought to elucidate how TECPR2-dependent secretory defects affect the molecular composition of lysosomes. Thereto, C-terminally HA- or FLAG-tagged TMEM192 was stably expressed in TECPR2 WT and MUT cells where tagged TMEM192 largely colocalized with lysosomes (Fig. 5a–c). Following lysosome immunoprecipitation (LysoIP)[22] (Supplementary Fig. 7a), lysosomal proteins were precipitated, digested with trypsin and analyzed by MS (Fig. 5d). Overall, we identified 570 lysosome-specific proteins out of a total of 3377 proteins (Supplementary Fig. 7b, Supplementary Data 6) with high reproducibility across all samples (Supplementary Fig. 7c), partial overlap between TECPR2 conditions (Supplementary Fig. 7d) and prominent lysosome-associated GO annotations (Supplementary Fig. 7e), confirming efficient and specific enrichment of lysosomes. By comparing TECPR2 WT and MUT cells, 100 and 123 proteins were found significantly decreased and increased, respectively, upon loss of full-length TECPR2 (Fig. 5e, f). Importantly, more than 65% of these altered lysosomal proteins did not show any overt changes in their total abundance (Fig. 5g, Supplementary Data 2). GO term analysis of this subpopulation revealed exclusive enrichment of autophagy-, intracellular transport- and endosome-related terms in TECPR2 MUT cells, while extracellular, cell-junction and -adherens clusters were only seen in TECPR2 WT cells (Fig. 5h). Using LysoIP coupled to immunoblotting as well as confocal microscopy, we validated the detected changes for a broad range of lysosomal proteins including the lipoprotein receptor LRP1 and the metalloprotease ADAMTS1 which both showed decreased lysosomal protein levels (Fig. 5i, j). Remarkably, almost all LAMTOR components of the Ragulator complex were enriched in TECPR2 MUT cells while Raptor (RPTOR) and MTOR were decreased (Fig. 5i, j), potentially indicating disturbed mTORC1 signaling that might contribute to the reported impairment of autophagosome formation upon depletion of TECPR2[5]. Moreover, in agreement with an observed altered lysosomal morphology[6], we detected increased lysosomal levels of LAMP1 and LAMP2 upon loss of full-length TECPR2 (Fig. 5a, i). Additionally, components of HOPS (VPS11) and the retromer cargo-selective complex (CSC) (VPS26A, VPS35) as well as the endosomal microautophagy receptor TOLLIP[17] were confirmed to increase on lysosomes in TECPR2 MUT cells (Fig. 5i), indicating potential defects in endosomal-lysosomal fusion or delivery. Comparative analysis of lysosomal proteins identified in this study with previously described autophagy-dependent lysosomal content[17] showed some overlap but no clear tendency for an increase or decrease of these proteins in TECPR2 MUT cells (Supplementary Fig. 7f), suggesting that autophagy is not induced despite an increase in ER stress in cells lacking TECPR2 (Supplementary Fig. 7g). Since ER stress is a known inducer of ER-phagy, we assessed the effect of TECPR2 deficiency on ER-phagy using automated live-cell imaging and two different ER-phagy reporters (Supplementary Fig. 7h). Whilst basal ER-phagy remained unaffected, the overall capacity to perform ER-phagy upon stimulation was considerably decreased. In accordance with a recent publication linking ERES to ER-phagy[23], the observed decrease in ER-phagy capacity might directly arise from the disruption of ERES upon TECPR2 deficiency. In summary, the membrane composition of lysosomes and their content profoundly changes in our disease-mimicking TECPR2 cell model, possibly as a downstream effect of disrupted trafficking through the secretory pathway towards lysosomes.

## Full-length but not truncated TECPR2 associates with ER-Golgi interface components

Since the majority of trafficking and sorting defect that we mapped in our HSAN9-mimicking cell model are consistent with a loss of full-length TECPR2 function within the early secretory pathway, we sought to substantiate this notion with a comparative interactome analysis of full-length and truncated TECPR2. Thereto, we reconstituted TECPR2 MUT cells with HA-TECPR2 WT or L440Rfs (Fig. 2a) and subjected them to immunoprecipitation (IP) followed by mass spectrometry (MS) (Fig. 6a) to identify binding partners that exclusively bind the full-length protein. Empty TECPR2 WT and TECPR2 MUT cells as well as TECPR2 MUT cells expressing HA-tagged FIP200/RB1CC1 as an unrelated bait protein were used as controls (Supplementary Fig. 8a). Stringent filtering, label-free quantification and statistical analysis (Supplementary Fig. 8a) of four biological replicates measured in technical duplicates with high reproducibility (Supplementary Fig. 8b) unveiled 8 respective 84 candidate interacting proteins for L440Rfs and WT TECPR2 (Fig. 6b, Supplementary Data 7 and Supplementary Data 8). Intriguingly, the latter pool contained a number of candidates which were also statistically enriched when TECPR2 WT was compared to empty background or FIP200 (Supplementary Fig. 8c, d, Supplementary Data 7 and Supplementary Data 8) and that prominently featured ER-, Golgi- or ER-to-Golgi transport-related functions. Among these were known interactors of the human ATG8 family (GABARAP, GABARAPL1, GABARAPL2) and potential new binding partners such as the COPII components SEC24C and SEC23A, the TRAPP subunits TRAPPC4, TRAPPC8 and TRAPPC11, the Golgi-localized casein kinase I isoform delta (CSNK1D) as well as the ER membrane proteins VAPA and SEC61B (Fig. 6b). Validation experiments not only confirmed that these candidates specifically bind to full-length and not disease-truncated TECPR2 at exogenous and endogenous levels but also indicated that some of these associations may take place in the context of higher-order protein assemblies such as the VAPA/B heterodimer, the TRAPP complex and the COPII coat (Fig. 6c, d). This notion is based on the observation that TECPR2 binds additional subunits of these assemblies (VAPB, TRAPPC9, SEC24D, SEC13) which were not detected or below the stringency threshold in the proteomic analysis. To further

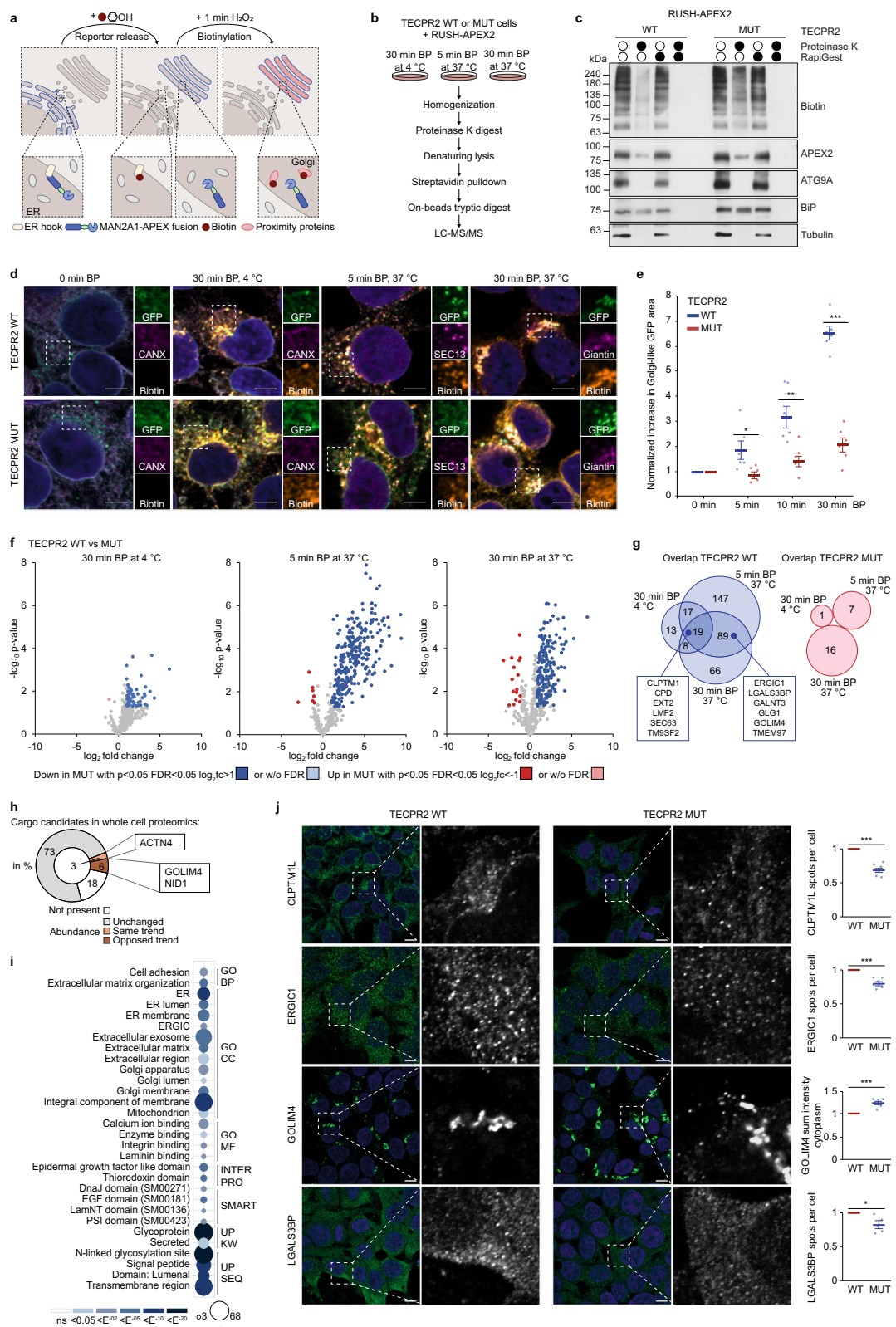

corroborate our findings, we performed colocalization studies and observed that the subcellular distribution of the outer COPII coat protein SEC31A and the VAPA dimer partner VAPB partially overlapped with that of WT but not L440Rfs TECPR2 (Fig. 6e). Since all of these binding partners are dynamically or constitutively localized to ER or Golgi structures, we next asked whether TECPR2 is likewise associated with these membranes. Firstly, we N-terminally tagged TECPR2 WT and

L440Rfs with myc-APEX2. Upon confirming that both fusion proteins were equally expressed and biotinylation competent in 293T cells (Supplementary Fig. 8e), we took advantage of APEX2 as a genetic tag for electron microscopy to obtain an ultrastructural view on TECPR2's proximity. Following post-fixation 3,3-diaminobenzidine (DAB) labeling and $H_2O_2$ pulsing, we detected defined contrast in the vicinity of vesicular structures upon expression of WT TECPR2 (Fig. 7a). In

**Fig. 3 | Mutant TECPR2 affects ER exit and trafficking of COPII cargo. a** Scheme of RUSH (MAN2A1-SBP-GFP) coupling to APEX2-mediated biotinylation in proteinase K protected trafficking compartments. **b** RUSH conditions and proteomics workflow. **c** Homogenates from TECPR2 WT and MUT cells expressing RUSH-APEX2 were left untreated or incubated with proteinase K, RapiGest or both. **d** ER-to-Golgi trafficking and biotinylation of RUSH-APEX2. TECPR2 WT and MUT cells expressing RUSH-APEX2 were treated with biotin-phenol (BP) as indicated followed by fixation, immunostaining and analysis by confocal microscopy. Scale bars 10 μm. **e** Quantification of RUSH-APEX2 trafficking by rationing the increase of the Golgi-like GFP-positive area to the total GFP-positive area (normalized to 0 min BP, two-sided t-test analysis, *n* = 6 independent experiments, error bars represent mean ± SEM, *p* values = 0.03261, 0.00637 and <0.00001). **f** Volcano plots of protease-protected, biotinylated proteins altered upon expression of truncated TECPR2 at indicated trafficking conditions. Dark colors highlight class I and light colors class II proteins (two-sided t-test, *n* = 4 independent experiments). **g** Venn diagram of class I and class II proteins similarly altered in different trafficking conditions in RUSH-APEX2 expressing TECPR2 WT and MUT cells. **h** Total abundance of altered RUSH-APEX2 candidates (in percent) in the whole cell proteomic analysis of TECPR2 WT and MUT cells. Orange and brown depict same and opposing trend of changes in RUSH-APEX2 and whole cell proteomics; gray, not changed; white, not present. Selected examples are highlighted. **i** GO term enrichment of proteins similarly altered at least between 5 min and 30 min BP at 37 °C. Dot size correlates to number of proteins, dot color to term enrichment (p-value). GO BP, biological process; GO CC, cellular compartment; GO MF, molecular function; UP KW, uniport keyword; UP SEQ, uniport sequence feature. **j** TECPR2 WT and MUT cells were fixed and immunolabeled (left panel). Insets show area of magnification (right panel). Scale bars 10 μm. Quantification of trafficking features (normalized to WT, two-sided t-test analysis, *n* = 5 independent experiments for LGALS3BP (*p* value = 0.02717) and *n* = 6 independent experiments for CLPTM1L, ERGIC1 and GOLIM4 (*p* value = 0.00001, 0.00031 and 0.00003), error bars represent mean ± SEM). Source data are provided as a Source Data file.

contrast, expression of the L440Rfs variant resulted in a more diffuse cytosolic staining with an additional signal in the nucleus (Fig. 7a). Secondly, we examined total membrane fractions of TECPR2 WT and MUT cells and detected a clear membrane association of endogenous full-length TECPR2 (Fig. 7b). Consistent with a recruitment of TECPR2 to the cytosolic side of these membranes, we observed no protection of TECPR2 from proteinase K treatment in cell homogenates (Supplementary Fig. 8f). While TECPR2 binding to ATG8 proteins and their lipidation was previously shown to be important for TECPR2's role in maintaining ERES and facilitating ER-to-Golgi transport, we unexpectedly found that inhibiting ATG8 lipidation did not alter general membrane association of TECPR2 (Fig. 7c). Consistently, endogenous full-length TECPR2 still associated with membranes in ATG8 hexa KO cells[24] which lack all six ATG8 family members (Fig. 7d). To further dissect TECPR2's association with the different membranous compartments, we performed multi-gradient differential centrifugation (Fig. 7e, f). In this analysis, endogenous full-length TECPR2 was found to be associated with fractions positive for ER, ERES and ERGIC markers (Fig. 7f). To further verify TECPR2's association with membranes of these compartments, we subjected TECPR2 MUT cells reconstituted with HA-TECPR2 WT or L440Rfs to homogenization without detergents followed by HA immune isolation. Intriguingly, we detected differential enrichment of the ER constituent VAPB, the cis-Golgi protein GM130 as well as the COPII carrier subunit SEC24D (Fig. 7g). Overall, these experiments show that full-length TECPR2 associates with several ER and ERGIC bound components of the early secretory pathway, thereby providing initial evidence that TECPR2's function within this pathway might go beyond the proposed stabilization of COPII components[5]. Moreover, a large portion of these associations was strongly reduced or completely lost when the C-terminal part of TECPR2 was lacking, strengthening the idea that a loss-of-function mechanism might be responsible for HSAN9.

## Discussion

In this study, we used several different spatial proteomics approaches to holistically determine defects in protein trafficking and sorting upon loss of full-length TECPR2. While TECPR2 has previously been linked to ER export[5], the functional consequences of defective TECPR2 for the secretory pathway remained largely elusive. Here, we identified proteins whose transport out of the ER to the Golgi and later compartments such as the lysosome, the PM and the extracellular space is altered in a full-length TECPR2-dependent manner. Among them were numerous proteins that can be linked to morphological and molecular alterations observed in recently established TECPR2-associated disease models[6,25] as well as factors associated with the development and maintenance of neuronal functions that might be related to clinical manifestations in HSAN9 patients[4]. In addition, we uncovered several new interactions of TECPR2 with proteins localized at or between the

ER and/or Golgi which are lost in disease-mimicking conditions, indicating that the interface between these two compartments might be the operation field of TECPR2 and not only individual COPII carrier components as previously reported (Fig. 7h). Together, this data provides a rich resource for further dissecting the molecular basis of HSAN9 and possibly related diseases and for developing biomarkers to potentially monitor TECPR2 deficiency in advanced cell and animal models.

Across the secretory pathway, we detected distinct changes in the protein composition of its different compartments in cells lacking TECPR2. Firstly, ERES were found to be disintegrated. Numerous proteins with roles in formation and regulation of COPII coats and ERES as well as factors involved in targeting COPII-coated vesicles to the Golgi were altered in their spatial distribution, indicating a general remodeling of the interfaces within the ER-ERGIC-Golgi axis. Consistent with this notion, a recent study observed the formation of Golgi-ERES units that depend on GOLGA2 and USO1 and are involved in efficient secretion in differentiating muscle cells[26]. Intriguingly, several proteins whose proximity to ERES depend on TECPR2 such as TFG, SEC31A, SPG20, WDR62 and GOLGA2 are themselves involved in neurological diseases[27-32], raising the possibility that their loss or gain of function contributes to the complex phenotypes observed in HSAN9 patients. Secondly and a likely consequence of defective secretory cargo egress from the ER, loss of full-length TECPR2 led to upregulation of the ER stress protein BiP, as well as changes in late secretory pathway compartments including lysosome, PM and extracellular space. For example, proteins associated with axon and neuronal development (PLEXINs) as well as cell-cell communication and adhesion (NCAM1, TNC, CRLF1) were decreased at the cell surface and in the extracellular space. These candidates might provide a functional link between an altered secretory pathway and phenotypes observed in HSAN9 patients. The validation of a number of these factors as TECPR2-dependent secretory cargo shows their potential to serve as biomarkers for TECPR2 function and HSAN9.

Besides, we observed extensive associations of TECPR2 with membranes of early secretory pathway compartments whose composition changed in the absence of TECPR2. At these surfaces, we identified several new interactions with distinct trafficking components including ER-and cis Golgi-associated proteins (Fig. 6g). Intriguingly, membrane recruitment of TECPR2 was not dependent on ATG8 proteins, indicating that these well-established TECPR2 binding partners might serve different - possible non-canonical - functions in conjunction with TECPR2[5].

TECPR2's domain architecture indeed provides arguments for potential other membrane association mechanisms and a general scaffolding function. Its N-terminal WD40 and C-terminal TECPR motifs are predicted to assemble into seven-bladed beta-propeller structures which can provide a platform for protein complex

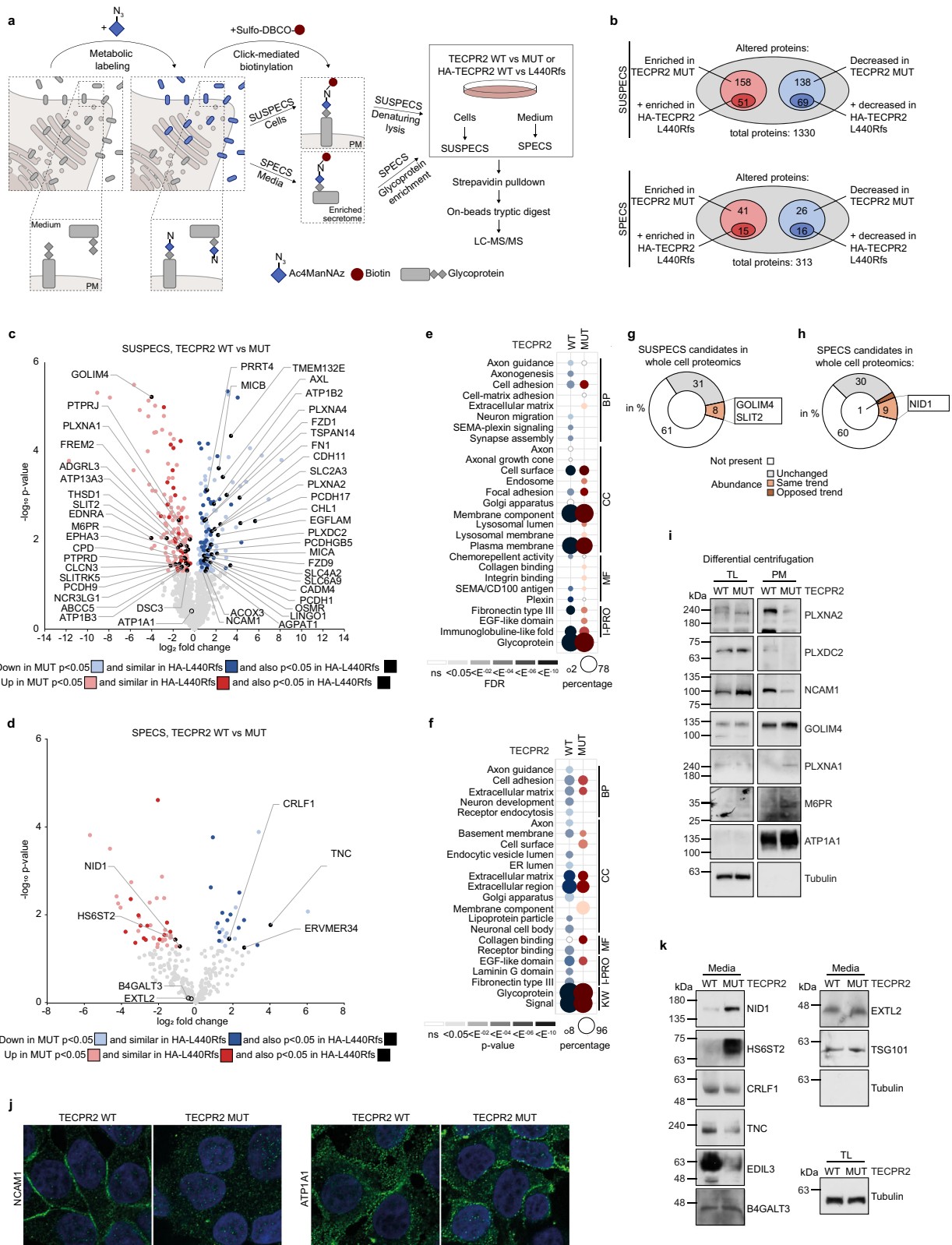

assembly[33]. The importance of both of these structures is supported by the fact that HSAN9 mutations tend to cluster in TECPR2 regions encoding these folds[4]. While the majority of these mutations are missense mutations, the assumption is that the resulting substitutions will likely disturb the beta-propeller either at the N- or C-terminus, thereby leading to destabilization and degradation of the whole protein. Our work indicates that the TECPR motifs are critical since expression of

the L440Rfs*19 mutant lacking this part of the protein cannot sustain full-length TECPR2 function with regard to facilitate trafficking thought the secretory pathway or to associate with ER and Golgi proteins. Although our data point to a general ER export defect, our RUSH-APEX2 method proved to be a useful tool to map COPII cargo. Expanding this approach to other RUSH reporters might help to further dissect specific effects on subpopulations of ERES/ERGIC carriers.

**Fig. 4 | Cell surface proteome and secretome composition changes upon expression of mutant TECPR2. a** Workflow of click-chemistry biotinylation of metabolically labeled glycoproteins from the PM (SUSPECS) and media (SPECS). **b** SUSPECS and SPECS analysis of TECPR2 WT and MUT cells as well as of TECPR2 MUT 293T cells reconstituted with TECPR2 WT or L440Rfs. Proteins enriched over non-biotinylated background are shown in gray, increased or decreased proteins upon TECPR2 MUT or TECPR2 L440Rfs expression are highlighted in red and blue, respectively. Light colors represent changes detected when comparing TECPR2 WT vs. MUT cells; dark colors represent proteins similarly affected in TECPR2 WT vs. L440Rfs. Volcano plot of cell surface proteome (**c**) and secretome (**d**) alterations upon expression of mutant TECPR2 at endogenous levels. Significantly altered proteins (two-sided t-test, $n = 4$ independent experiments) are grouped as class III (light color), class II (dark color) or class I (black and annotated). GO term enrichment of SUSPECS (**e**) and SPECS (**f**) proteins found in TECPR2 WT and MUT cells. Dot size correlates to number of proteins, dot color to term enrichment (FDR). Total abundance of altered SUSPECS (**g**) and SPECS (**h**) candidates (in percent) in the whole cell proteomic analysis of TECPR2 WT and MUT cells. Orange and brown depict same and opposing trends of changes in SUSPECS/SPECS and whole cell proteomics; gray, not changed; white, not present. Selected examples are highlighted. **i** Immunoblot of PM fractions obtained by differential centrifugation of homogenates from TECPR2 WT and MUT cells. ATP1A1 and tubulin served as loading controls. **j** TECPR2 WT and MUT 293T cells were fixed and immunolabeled. Scale bars 10 μm. **k** Media from TECPR2 WT and MUT cells were subjected to size-exclusion filtration and lectin-based immunoprecipitation followed by immunoblotting. B4GALT3, EXTL2, TSG101 and tubulin served as loading controls. Source data are provided as a Source Data file.

As ERES contribute to autophagosome biogenesis[14,34], it is likely that secretory and autophagic defects observed in TECPR2 MUT cells are intertwined. This notion is supported by the presence of VAPA and VAPB in the TECPR2 interactome which were both shown be involved in autophagosome formation[35]. Disintegrated ERES and reduced formation of COPII-coated carriers might directly alter the ERGIC, resulting in reduced membrane supply for nascent phagophores. Additionally, TECPR2 deficiency was found to reduce the capacity of ER-phagy, which was previously shown to be supported by functional ERES[23]. A recent study described a hybrid pre-autophagosomal structure that is formed by fusion of Golgi and endosomal membranes[36]. As TECPR2 associates with both compartments, a similar mechanism linking autophagy and secretion might be compromised in the context of HSNA9. Furthermore, loss of full-length TECPR2 increased the association of HOPS and LAMTOR complexes with lysosomes while MTOR and RPTOR were reduced. However, it remains to be tested whether these changes are direct consequences of the trafficking and protein sorting defects caused by a loss of full-length TECPR2 function within the early secretory pathway. As disturbances in endo-lysosomal trafficking also affect unconventional protein secretion, studies focusing on these aspects, e.g., exosomal secretion, might help to further define the effects of TECPR2 deficiency.

Overall, our systematic proteomic analysis of different steps in the secretory pathway led to the identification of a specific set of proteins whose sorting or binding is altered in a manner depending on full-length TECPR2. Importantly, these molecules are potential biomarkers for studying functional consequences of TECPR2 deficiency in HSAN9 animal models and advanced patient-derived cell systems. Moreover, our approach may serve as a blueprint to unbiasedly investigate trafficking and secretory defects in other rare neurodegenerative and -developmental diseases.

## Methods
A summary of all used reagents, antibodies, hardware and tools can be found in Supplementary Data 9, including unique identifiers, order numbers, version information and other relevant information.

### Cell culture
HeLa (ATCC, #CCL-2), SH-SY5Y (ATCC, #CRL-2266), fibroblasts (patient-derived) and 293T (ATCC, #CRL-3216) cells were grown in Dulbecco's modified Eagle medium (DMEM GIBCO, #61965-026) supplemented with 10% fetal bovine serum (FBS) and 1% sodium pyruvate. Stable cell lines were grown in medium supplemented with puromycin (2 mg/ml Sigma, #P8833).

### Cloning
All cloning was performed using the Gateway cloning system. ORFs were transferred into the entry vector pDONOR233 after being flanked with attB sites. Once in pDONOR233, ORFs were cloned into pHAGE myc-APEX2[17], pHAGE HA-FLAG and pHAGE GFP vectors using

homologous recombination. An overview of generated and used recombinant DNA can be found in Supplementary Data 9.

### Cell line generation
293T and HeLa cells expressing L440Rfs-mimicking endogenous TECPR2 (referred to as TECPR2 MUT cells) were generated by CRISPR/Cas9-mediated genome editing. GRNAs were designed using the Broad Institute GPP gRNA designer tool, directed towards exon 8 and resulted in a G421Rfs*29 (in 293T) or a D417Efs*15 (in HeLa) truncation as confirmed by sequencing. Cell lines stably expressing tagged TECPR2, MAN2A1-RUSH-APEX2, tagged TMEM192 or the ER-phagy reporters ssRFP-GFP-KDEL and mKeima-RAMP4 were generated by lentiviral transduction. 293T cells were transfected with lentiviral constructs using Lipofectamine (Invitrogen, ratio 2 μg DNA to 10 μl Lipofectamine 2000) and recipient HeLa and 293T cells were selected in medium supplemented with 2 μg/ml Puromycin or 15 μg/ml Blasticidin 48 h post-transduction, respectively. ER-phagy reporter expressing 293T cells were sorted for the same populations of RFP- and GFP-expressing cells after 3 passages. An overview of used gRNAs and plasmids can be found in Supplementary Data 9.

### Transfections and treatments
Transient transfections were performed using PEI (Polyethylenimine, Polysciences Europe GmbH) or Lipofectamine 2000 (Invitrogen) according to standard protocols for 12 h to 48 h. RNAi-mediated knockdown of TECPR2 was performed using Lipofectamine RNAiMAX (Invitrogen) according to manufacturer's protocol for 48 h. The following reagents were used as indicated: ATG7 Inhibitor (Takeda, 1 μM, 24 h), Biotin-Phenol (IrisBiotech, 500 μM, 5–30 min), $H_2O_2$ (Sigma, 1 mM, 1 min), Proteinase K (Roche, 30 μg/ml for 30 min), RAPIGest™ (Waters, 0.1%, 30 min), Triton X-100 (Merck, 0.2%, 30 min), PMSF (Sigma-Aldrich, 10 mM), Click-IT™ ManNAz Metabolic Glycoprotein Labeling Reagent (Thermo Scientific, 50 μM, 24 h) and Sulfo-DBCO-Biotin Conjugate (Jena Biosciences, 50 μM, 2 h). An overview of used siRNAs and plasmids can be found in Supplementary Data 9.

### Antibodies
All primary antibodies were used at a concentration of 1:1000 in 5% milk or 5% BSA in TBS supplemented with 0.5% Tween-20 for immunoblotting (IB) or 1:300–500 in 0.1% BSA in PBS for immunofluorescence (IF) unless otherwise stated. Secondary antibodies were used at a concentration of 1:10,000 in 5% milk in TBS-Tween for IB and 1:600 in 0.1% BSA-PBS for IF. All antibodies were validated by the respective supplier, checked for proper protein detection by immunoblotting and immunofluorescence analysis and are listed with unique RRID identifiers in Supplementary Data 9. The following primary antibodies were used:ADAMTS1 (Abcam, #ab39194), APEX2 (IgG2A) (custom made by Regina Feederle, HZM München), ATP1A1 (Abcam, #ab7671), B4GALT3 (Proteintech, #11041-1-AP), BAG2 (Biomol, #A304-751A), beta-ACTIN (Sigma,#A1978), BIOTIN (Pierce, #31852), BiP (Cell Signaling, #3177), CALNEXIN (Abcam, #ab22595),

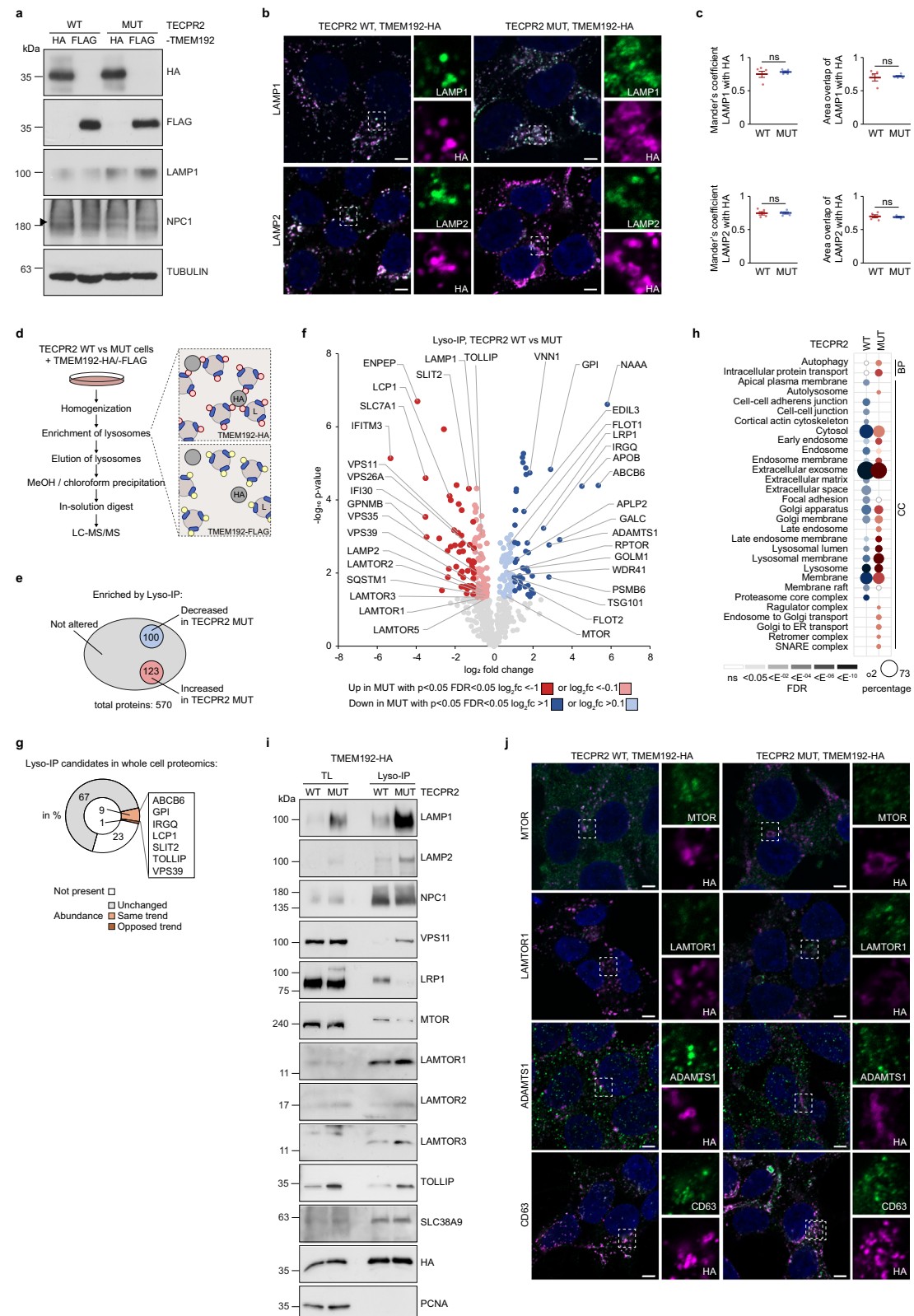

Calreticulin (santa cruz, #sc-6467), CD63 (abcam, #ab59479), CLPTM1L (Sigma, #HPA014791), c-myc (Bethyl, #A190-104A), CRLF1 (Novus, #NBP1-85606), DTNBP1 (Bethyl, #A303-360A), EDIL3 (Abcam, #ab190692), ERGIC1 (Proteintech, #16108-1-AP), ERGIC53 (Santa Cruz, #sc398777), EXTL2 (Abcam, #ab168391), Flag M2 (Cell Signaling, #2368), GABARAP(Abcam,#109364), Giantin (Biolegend, #924302), GM130 (Abcam, #ab52649), GOLIM4 (Abcam, #ab28049), HA.11 Clone 16B2 (Covance/Biolegend, #MMS-101P/#901501), HS6ST2 (Abcam, #ab122220), LAMP1 (IF 1:100, Abcam/DSHB, #ab24170/H4A3), LAMP2 (IF 1:100, Abcam, #ab25631), LAMTOR1 (Cell Signaling, #8975S), LAMTOR2 (Cell Signaling, #8145); LAMTOR3 (Cell Signaling, #8168), LC3B (Cell Signaling, #2775S), LGALS3BP (Proteintech, #10281-1-AP), LRP1 (Abcam, #ab92544), M6PR (Abcam, #ab2733), MTOR (Cell Signaling, #2983), myc 9E10 (custom made by Regina Feederle, HZM),

**Fig. 5 | TECPR2 deficiency impacts the composition of lysosomes. a** TECPR2 WT and MUT cells expressing C-terminally 3×HA or 2×FLAG-tagged TMEM192 were lysed and immunoblotted. Tubulin served as loading control. **b** TECPR2 WT and MUT cells expressing C-terminally 3×HA-tagged TMEM192 were fixed and immunolabeled. LAMP1 and LAMP2 served as lysosomal marker. Scale bar 10 μm. **c** Quantification of colocalization of LAMP1 and LAMP2 with 3×HA-tagged TMEM192 between TECPR2 WT and MUT cells per statistical two-sided t-test analysis of normalized overlap measures (n = 4 independent experiments). Error bars represent mean ± SEM, p values = 0.75034 and 0.77539 (for LAMP1) and 0.91922 and 0.91032 (LAMP2). **d** Scheme of LysoIP workflow. **e** Venn graph-like summary of proteins enriched by TMEM192-3×HA over TMEM192-2×FLAG background binding (gray circle) and of proteins increased (red circle) and decreased (blue circle) in TECPR2 WT and MUT cells, respectively. **f** Volcano plot of proteins identified by LysoIP in TECPR2 WT and MUT cells. Significantly altered proteins (two-sided t-test,

n = 4 independent experiments) are grouped as class I (dark color) or class II (light color). Known lysosomal components and selected candidates are highlighted. **g** Total abundance of altered LysoIP candidates (in percent) in the whole cell proteomic analysis of TECPR2 WT and MUT cells. Orange and brown depict same and opposing trends of changes in LysoIP and whole cell proteomics; gray, not changed; white, not present. Selected examples are highlighted. **h** GO term enrichment of altered LysoIP proteins in TECPR2 WT and MUT cells. Dot size correlates to number of proteins, dot color to term enrichment (FDR). **i** Homogenates from TECPR2 WT and MUT cells expressing TMEM192-3×HA were subjected to LysoIP and immunoblotting. SLC38A9, HA and PCNA served as loading controls. **j** TECPR2 WT and MUT cells expressing TMEM192-3×HA were fixed and immunolabeled. Insets show magnifications. CD63 served as an endolysosomal marker. Scale bar 10 μm. Source data are provided as a Source Data file.

NCAM1 (WB: 1:500, IF 1:100, Merck, #AB5032), NEK9 (Abcam, #ab138488), NID1 (Invitrogen, #PA5-99666), NPC1 (WB: 1:500, Abcam, #ab134113), PCNA (Santa Cruz, #sc-7907), PLXDC2 (Novus, #NBP1-76858), PLXNA1 (R&D, #AF4309), PLXNA2 (Abcam, #ab39357), RAB5C (Sigma,#HPA003426), SEC12 (Novus, #NBP1-87056), SEC13 (Novus, #AF9055-100), SEC24C (Abcam, #ab122633), SEC24D (Cell Signaling, #14687), SEC31A (BD, #612351), SLC38A9 (Abcam, #ab81687), SPG20 (Proteintech, #13791-1-AP), TECPR2 (Christian Behrends, custom made), TNC (Abcam, #ab108930), TOLLIP (Abcam, #ab187198), TOMM40 (Abcam, #ab185543), TRAPPC11 (Sigma, #HPA045427), TRAPPC8 (Sigma, #HPA041107), TRAPPC9 (Proteintech, #16014-1-AP), TSG101 (Abcam, #ab30871), TUBULIN (Abcam, #ab7291), VAPA (Sigma, # HPA009174), VAPB (Sigma, # HPA013144), VCP (Bethyl, #A300-588A), VPS11 (Abcam, #ab125083). The following secondary antibodies were used: anti-goat-HRP (Promega, #V8051), anti-mouse-HRP (Promega, #W402B), anti-rabbit-HRP (Promega, #W4011), anti-rat-HRP (Sigma, #A-9037), Donkey anti-goat-488 (Life Technologies, #A11055), Donkey anti-mouse-488 (Life Technologies, #A21202), Donkey anti-rabbit-488 (Life Technologies, #A21206), Goat anti-rabbit-488 (Life Technologies, #A11034), Goat anti-mouse-488 (Life Technologies, #A11001), Donkey anti-goat-555 (Life Technologies, #A21432), Donkey anti-mouse-555 (Life Technologies, #A32773), Donkey anti-rabbit-555 (Life Technologies, #A31572), Goat anti-mouse-555 (Life Technologies, #A21424), Donkey anti-goat-640 (Life Technologies, #A32849).

## Immunoblotting

For total lysates, cells were lysed in RIPA-buffer (50 mM Tris, 150 mM NaCl, 0.1% SDS, 0.5% sodium deoxycholate, 1% NP-40, 1× protease inhibitors (Roche) and 1× PhosStop (Roche)) or MCL-buffer (50 mM Tris, 150 mM NaCl, 1% NP-40, 1× protease inhibitors l(Roche) and 1× PhosStop (Roche)) for 30 min on ice. After clearance by centrifugation at 20,000 × g for 10 min at 4 °C, lysates were boiled in sample buffer (200 mM Tris-HCL, 6% SDS, 20% Glycerol, 0.1 g/ml DTT and 0.01 mg Bromophenol Blue). Proteins were separated by SDS-PAGE and transferred onto nitrocellulose membranes. Membranes were blocked in 5% milk or 5% BSA in TBS supplemented with 0.1% Tween-20 (TBS-T) respectively. Membranes were incubated with primary antibodies overnight at 4 °C in blocking buffer, washed with TBS-T, incubated with secondary HRP-conjugated antibodies for 1 h at room temperature, washed again with TBS-T and immediately analyzed by enhanced chemiluminescence.

## Immunofluorescence

293T cells were grown on lysine (Sigma)-coated coverslips, HeLa and SH-SY5Y on uncoated coverslips. Cells were washed with DPBS (Gibco), fixed with 4% paraformaldehyde (Santa Cruz) for 10 min and subsequently permeabilized with 0.5% TritonX-100 (Merck) for 10 min. For plasma membrane staining, permeabilization was omitted. Before sequential staining with primary and ALEXA-coupled secondary

antibodies, cells were blocked with 1% BSA for 1 h at room temperature. All antibodies were stained for 1 h at room temperature in the dark. Coverslips were mounted in Prolong Gold supplemented with DAPI (Invitrogen) and imaged on a LSM800 confocal microscope (Zeiss) with an 63× oil-immersion objective. Images were processed with Zen blue (v. 2.5) (Zeiss). For images taken at the OPERA high-content screening system (PerkinElmer), cells were grown in lysine-coated 96 well-plates.

## Proximity labeling of ERES components

Cells transiently expressing APEX2-ERES/COPII chimeras were incubated with DMEM supplemented with 500 μM biotin-phenol (IrisBiotech) at 37 °C for 30 min before addition of 1 mM $H_2O_2$ at room temperature for 60 s to trigger peroxidase activity. Biotinylation was immediately quenched by two washes in quencher solution (1 mM sodium azide, 10 mM sodium ascorbate and 5 mM Trolox in DPBS)[37]. Cells were washed once in DPBS, harvested and cell counts equally adjusted. Cell pellets were lysed with freshly prepared quenching-RIPA buffer (50 mM Tris, 150 mM NaCl, 0.1% SDS, 0.5% sodium deoxycholate, 1% Triton X-100, 1× protease inhibitors (Roche), 1× PhosStop (Roche), 1 mM sodium azide, 10 mM sodium ascorbate and 1 mM Trolox) on ice for 30 min. Cell lysates were cleared by centrifugation at 20,000 × g and supernatant were incubated over-night with pre-equilibrated streptavidin agarose (Sigma). Beads were subsequently washed 3× with quenching-RIPA buffer, 3× with freshly prepared 3 M Urea buffer (Urea in 50 mM ammonium bicarbonate) and suspended in a defined volume of 3 M Urea buffer. Samples were reduced with 5 mM TCEP (Sigma) at 55 °C for 30 min, alkylated with 10 mM IAA (Sigma) at room temperature for 20 min and quenched with 20 mM DTT (Sigma). Samples were washed with 2 M Urea buffer (Urea in 50 mM ammonium bicarbonate), suspended in 50 μl of 2 M Urea buffer and digested with 1 μg trypsin per sample at 37 °C overnight. Peptides were collected by pooling the supernatant with two 50 μl 2 M Urea buffer washes, immediately acidified with 1% trifluoroacetic acid and concentrated by vacuum centrifugation[38]. Digested peptides were desalted on custom-made C18 stage tips and solved in 0.1% formic acid. For immunoblotting, beads were washed 3× with quenching-RIPA buffer and boiled in sample buffer (200 mM Tris-HCL, 6% SDS, 20% Glycerol, 0.1 g/ml DTT and 0.01 mg Bromophenol Blue) supplemented with 30 mM biotin.

## COPII Cargo profiling

Samples were processed essentially as described above except for biotin-phenol and proteinase K treatment. Cells were incubated with DMEM supplemented with 500 μM biotin-phenol (IrisBiotech) for 0 min, 5 min or 30 min at 4 °C or 37 °C. Subsequently, biotinylation was triggered with 1 mM $H_2O_2$ at room temperature or on ice for 60 s. For proteinase K digests, cells were suspended in cold homogenization buffer I (10 mM KCl, 1.5 mM MgCl2, 10 mM HEPES-KOH and 1 mM DTT, pH 7.5) and incubated shaking for 20 min at 4 °C. Cells were

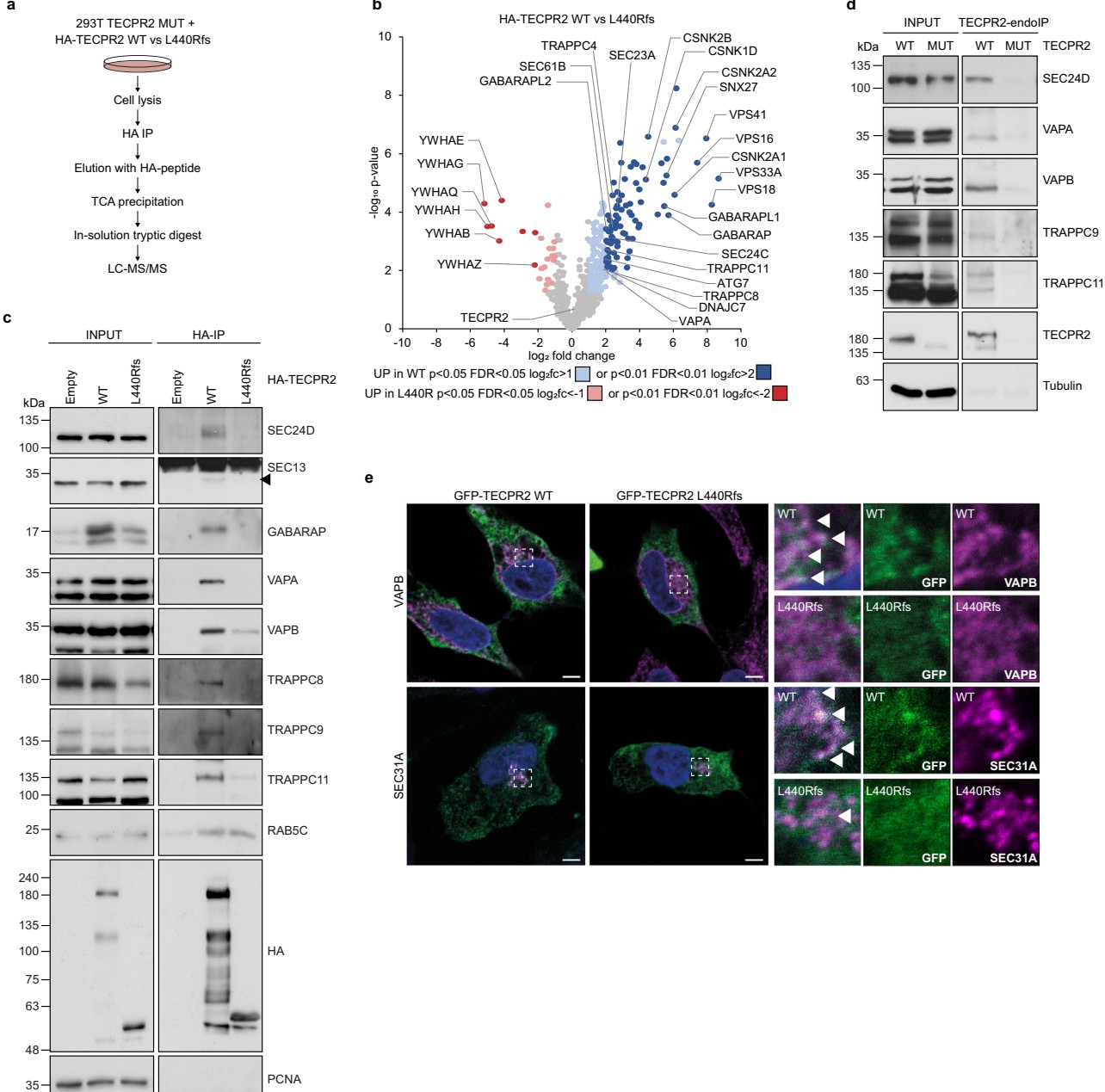

**Fig. 6 | Full-length TECPR2 associates with ER-Golgi interface components. a** IP-MS workflow. **b** Volcano plot of candidate interacting proteins identified from HA-TECPR2 WT or L440Rfs expressing cells. Candidates specific for TECPR2 WT and L440Rfs (two-sided t-test, $n = 4$ independent experiments) are colored blue and red, respectively. **c** Lysates from empty and HA-TECPR2 WT or L440Rfs expressing TECPR2 MUT cells were subjected to HA-IP followed by immunoblotting.

**d** Endogenous TECPR2 was immunoprecipitated from TECPR2 WT and MUT cell lysates followed by immunoblotting. Tubulin served as loading control. **e** HeLa cells transiently transfected with GFP-TECPR2 WT or L440Rfs were fixed and immuno-labeled. Insets show magnification of boxed areas. Scale bars 10 μm. Source data are provided as a Source Data file.

subsequently dounced utilizing a tight-fitting pestle on ice. Homogenates were mixed with cold homogenization buffer II (375 mM KCl, 22.5 mM MgCl2, 220 mM HEPES-KOH and 0.5 mM DTT, pH 7.5) at a ratio 1:5 (homogenization buffer I:II), centrifuged at $600 \times g$ for 10 min and supernatants incubated with proteinase K. For mass spectrometry, samples were incubated at 37 °C for 1 h with 100 mg/ml proteinase K or 0.1% RAPIGest™ as a control. For immunoblotting, samples were incubated at room-temperature for 30 min with 30 mg/ml proteinase K or 0.1% RAPIGest™ as a control. Digestion was stopped by 10 mM PMSF, samples were centrifuged at $17,000 \times g$ for 15 min and supernatants collected for lysis.

## SUSPECS

Glycoproteins were labeled in living cells using medium supplemented with 50 μM Click-IT™ ManNAz Metabolic Glycoprotein Labeling Reagent (Thermo Scientific) for 24 h. Cells were incubated with 100 μM Sulfo-DBCO-biotin (Jena biosciences) in PBS at 4 °C for 2 h to capture labeled glycoproteins with biotin[18]. Cells were extensively washed, harvested and samples adjusted to equal cell counts. Samples were similarly processed essentially as described above except that RIPA buffer (50 mM Tris, 150 mM NaCl, 0.1% SDS, 0.5% sodium deoxycholate, 1%Triton X-100, 1× protease inhibitors (Roche), 1× PhosStop (Roche)) was used for lysis.

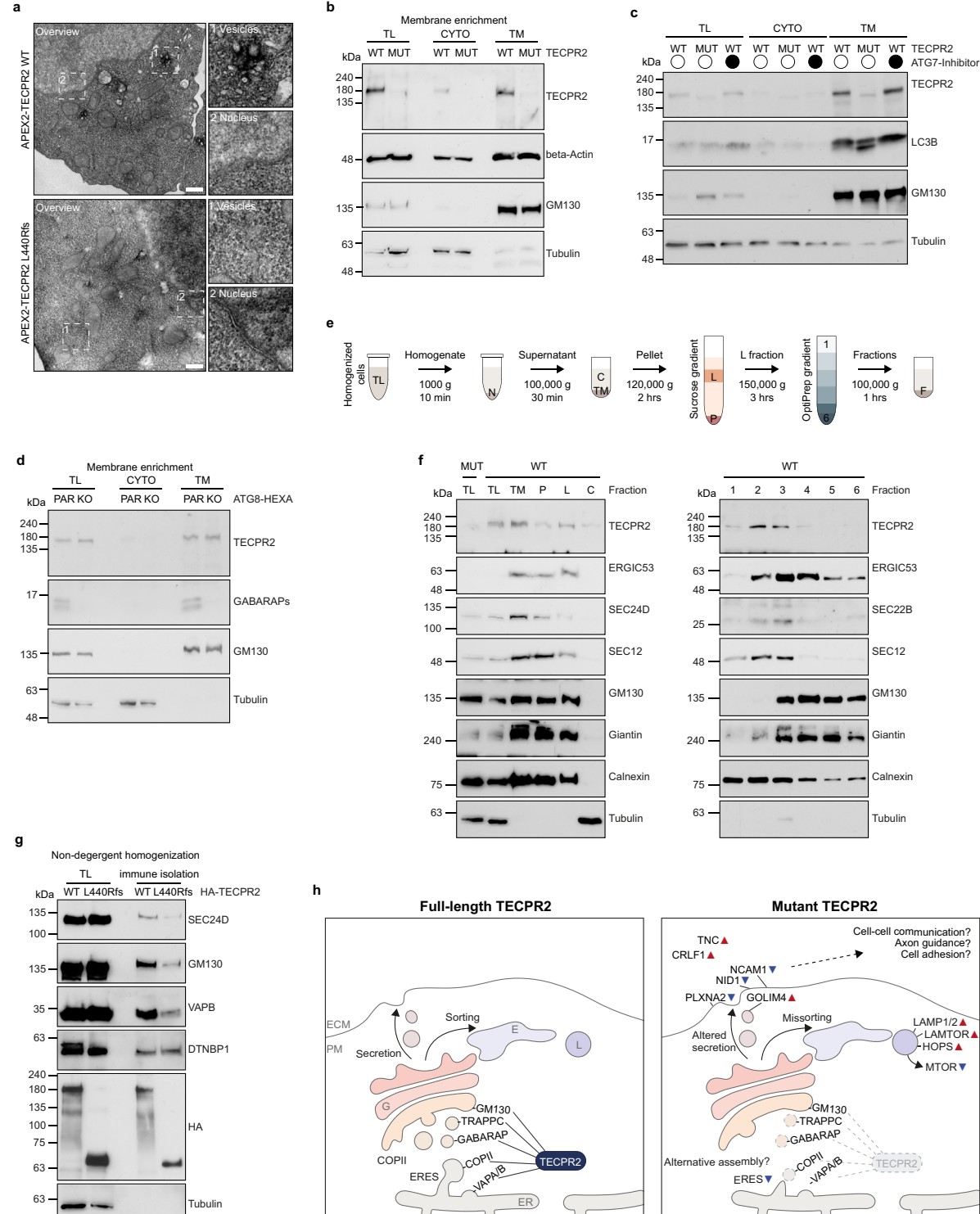

**SPECS**

Glycoproteins were labeled using medium supplemented with 50 μM Click-IT™ ManNAz Metabolic Glycoprotein Labeling Reagent (Thermo Scientific) in living cells. Media were harvested after 48 h, sterile-filtered, supplemented with proteinase inhibitors (Roche) and volumes adjusted based on cell counts. Media were then concentrated to 0.5 ml by centrifugation through 30 kDa diafilter (Sartorius) at 4 °C and retentates washed twice with 15 ml cold DPBS supplemented with protease inhibitors (Roche). 100 μM Sulfo-DBCO-biotin (Jena

biosciences) was added and retentates filled up to 1 ml. After adjustment to pH > 7, samples were incubated overnight at 4 °C. Samples were washed 3× with 10 ml cold DPBS and final retentates transferred to low-binding tubes. 300 μl per sample Concanavalin A sepharose (Sigma) was equilibrated with Binding buffer (5 mM MgCl₂, 5 mM MnCl₂, 5 mM CaCl₂, 500 mM NaCl in 20 mM Tris-HCL pH 7.5) and added to each sample. Volumes were filled up to 1.3 ml and samples incubated at 4 °C for 2 h with gentle rotation. Beads were washed with Binding buffer and proteins eluted twice with 500 μl Elution buffer

**Fig. 7 | Full-length TECPR2 localizes to ER and ERGIC membranes. a** Electron micrographs of APEX2-TECPR2 WT or L440Rfs expressing cells. Before embedding and ultrathin sectioning, fixed samples were incubated with DAB and $H_2O_2$. Insets show magnification of representative nuclear and vesicular areas. Scale bars 200 nm. **b** TECPR2 WT and MUT cells were subjected to homogenization and differential centrifugation. Cytosolic (CYTO) and total membrane (TM) fractions were analyzed by immunoblotting. GM130 and tubulin served as loading controls. TECPR2 WT and MUT cells differentially treated with an ATG7 inhibitor (**c**) or ATG8-HEXA KO HeLa cells (**d**) were subjected to homogenization and differential centrifugation. Total lysate (TL), cytosolic (CYTO) and total membrane (TM) fractions were analyzed by immunoblotting. GM130, LC3B, GABARAP and tubulin served as controls. **e** Schematics of two-step differential centrifugation, resulting in total lysates (TL), nuclear debris (N), cytosol (C), total membranes (TM), light sucrose membranes (L), a sucrose membrane pellet (P) and six different OptiPrep membrane fractions (1–6). **f** TECPR2 WT cells were subjected to homogenization followed by two-step differential centrifugation and immunoblotting. **g** HA-TECPR2 WT or L440Rfs were immune-isolated from homogenates and selected association partners detected by immunoblotting. DTNBP1 served as loading control. HA-IP, immune-isolated membranes. **h** Working model of TECPR2's field of operations at the ER-Golgi interface and protein sorting defects caused by truncated TECPR2. Source data are provided as a Source Data file.

(500 mM Methyl-α-D-mannopyranoside(Sigma), 10 mM EDTA in 20 mM Tris-HCL pH 7.5) for 30 min at 4 °C[19]. Combined eluates were mixed with 1 ml 2% SDS in PBS and samples subsequently processed as described above (proximity labeling).

## PM enrichment

An PM extraction kit (Abcam, Ab65400) was used according to the manufacturer's guidelines. Cells were collected by scaping, homogenized in homogenization buffer (kit) with a douncer and sequentially centrifuged to separate cytosolic and total membrane proteins. PM proteins were separated from the total membrane protein fraction by phase-separation, supplemented with sample buffer (200 mM Tris-HCL, 6% SDS, 20% Glycerol, 0.1 g/ml DTT and 0.01 mg Bromophenol Blue) and boiled for 5 min.

## Lyso-IP

TMEM192-HA or TMEM192-FLAG were stably overexpressed in TECPR2 WT or KO 293T cells and samples were essentially generated as described before[22]. Cells were washed twice with PBS, harvested and subsequently suspended in cold Lyso-IP buffer (50 mM KCl, 100 mM $KH_2PO_4$ and 100 mM $K_2HPO_4$ pH 7.2, supplemented with protease inhibitors). Following homogenization with a tight-fitting pestle, homogenates were centrifuged at $1500 \times g$ for 10 min, supernatants were transferred to fresh tubes and pre-equilibrated HA-magnetic beads (Thermo Fisher) were added. After 1 h rotation at 4 °C, beads were washed thrice with salt buffer (LysoIP buffer supplemented with 300 mM NaCl) and lysosomal proteins were eluted. For immunoblotting, beads were boiled in 100 μl sample buffer (200 mM Tris-HCL, 6% SDS, 20% Glycerol, 0.1 g/ml DTT and 0.01 mg Bromophenol Blue) for 5 min and supernatants subjected for analysis. For mass spectrometry, lysosomes were eluted and disrupted by incubating beads in 150 μl Urea buffer (8 M Urea, 50 mM Tris pH8, 150 mM NaCl) for 30 min shaking at 4 °C and sonicated for 5 min. Eluates were reduced with 5 mM TCEP (Sigma) at 55 °C for 30 min, alkylated with 15 mM IAA (Sigma) at room temperature for 30 min and quenched with 10 mM DTT (Sigma) at room temperature for 15 min. Proteins were precipitated using methanol chloroform-precipitation by sequentially mixing samples with 600 μl methanol (Roth), 150 μl chloroform (Merck) and 450 μl MS-grade water, removing the hydrophilic phase, washing with 450 μl methanol and pelleting proteins by centrifugation at $14,000 \times g$ for 5 min. Methanol was completely removed by vacuum centrifugation and precipitated proteins reconstituted in 30 μl 50 mM ABC followed by tryptic digestion with 1 μg trypsin per sample at 37 °C overnight. Digestion was terminated by the addition of 30 μl 5% acetonitrile (Roth)/5% formic acid (Merck). Digested peptides were dried by vacuum centrifugation, resuspended in 30 μl 5% acetonitrile (Roth)/1% TFA (Fluka), desalted on custom-made C18 stage tips and reconstituted in 0.1% formic acid.

## TECPR2 interactome

For pulldown of TECPR2-associated membrane compartments, samples were processed as described above (Lyso-IP) using 293T cells expressing HA-tagged TECPR2 variants. For immunoprecipitation of

HA-tagged TECPR2, cell pellets were lysed in MCL-buffer (50 mM Tris pH 7.4, 150 mM NaCl, 0.5% NP40, 1× protease inhibitors (Roche) rotating at 4 °C for 30 min. Lysates were cleared through 0.45 μm PVDF membranes (Millipore) and protein concentrations adjusted accordingly by BCA. Pre-equilibrated HA agarose (Sigma) was added to lysates and samples were incubated overnight at 4 °C with gentle rotation. Beads were washed extensively with MLC-buffer and PBS before proteins were eluted 3× with 50 μl 250 μg/ml HA peptide (Sigma) (in PBS) at room temperature for 30 min. Eluates were pooled, acidified with 26 μl of 100% TCA, vortexed and incubated on ice for 30 min. Samples were spun at $20,000 \times g$ at 4 °C for 30 min, protein pellets covered immediately with ice-cold 10% TCA and spun again at $20,000 \times g$ at 4 °C for 30 min. Pellets were washed 3× with −20 °C cold acetone with interjacent spins at $20,000 \times g$ at 4 °C for 10 min. Acetone supernatants were aspirated carefully and protein pellets dried completely before being reconstituted in 50 μl 50 mM ABC/10% ACN pH 8.0. Proteins were digested with 0.5 μg trypsin per sample at 37 °C. After 4 h, 30 μl of 5% formic acid/5% ACN was added, peptide mixtures incubated at room temperature for 10 min and peptides dried by vacuum centrifugation. Dried peptides were resuspended in 30 μl 5% ACN (Roth)/1% TFA (Fluka), desalted on custom-made C18 stage tips and reconstituted in 0.1% formic acid. For immunoblotting, proteins were eluted by boiling washed beads in sample buffer (200 mM Tris-HCL, 6% SDS, 20% Glycerol, 0.1 g/ml DTT and 0.01 mg Bromophenol Blue) for 5 min. For immunoprecipitation of endogenous TECPR2, cells were lysed in MCL-buffer (50 mM Tris pH 7.4, 150 mM NaCl, 0.5% NP40, 1× protease inhibitors (Roche) on ice for 30 min. Lysates were filtered through 0.45 μm PVDF membranes (Millipore), protein concentrations adjusted accordingly by BCA and lysates pre-cleared with protein-G sepharose at 4 °C for 1 h. Sepharose was removed and lysates were incubated with anti-TECPR2 antibody overnight at 4 °C with gentle rotation. Protein-G sepharose was added for another 2 h, beads pelleted, washed extensively MLC-buffer and proteins eluted by boiling beads in sample buffer (200 mM Tris-HCL, 6% SDS, 20% Glycerol, 0.1 g/ml DTT and 0.01 mg Bromophenol Blue) for 5 min.

## Electron microscopy

293T cells expressing APEX2-tagged TECPR2 WT and L440Rfs were grown on aclar sheets (Science Services) and fixed with 2.5% glutaraldehyde (EM-grade, Science Services) in 0.1 M pH 7.4 sodium cacodylate buffer (CB) for 30 min on ice. Endogenous peroxidases were blocked with 20 mM glycine (Sigma) for 5 min on ice and cells washed in CB. Cells were saturated with freshly prepared 1× diaminobenzidine (DAB, in CB supplemented with 2 mM calcium chloride) for 5 min and APEX2 activity was triggered with 1× DAB supplemented with 10 mM $H_2O_2$ (Sigma) for 40 min on ice. Cells were washed with CB and subsequently postfixed and contrasted in reduced osmium (1.15% osmium tetroxide (Science Services) 1.5% potassium ferricyanide (Sigma)) for 30 min. After washes in CB and $H_2O_2$, cells were incubated in 0.5% aqueous uranylacetate (Science Services) over-night and dehydrated using a graded series of ice-cold ethanol-water composite. Cell monolayers were infiltrated in epon (Serva) and cured for 48 h at 60 °C. 50 nm ultrathin sections were generated on formvar-coated

copper grids (Plano). Sections were imaged using a JEM-1400 + (JEOL) equipped with a XF416 (TVIPS) and the EM-Menu software (TVIPS) and analyzed using ShotMeister (JEOL).

## Whole proteome analysis

For whole proteome analysis, cells were harvested by scraping, washed twice with PBS and lysed in Urea-buffer (9 M Urea, 50 mM Tris pH 8, 150 mM NaCl, 1× protease inhibitors (Roche) rotating at 4 °C for 30 min. After sonicating, lysates were cleared by centrifugation at 2500 g and the protein amount was measured and adjusted by BCA assay. Samples were reduced with 5 mM DTT (Sigma) at 55 °C for 30 min, alkylated with 14 mM IAA (Sigma) at room temperature for 30 min and quenched with 5 mM DTT (Sigma) for 15 min. Subsequently, samples were diluted 1:5 with 1 M Tris-HCl, pH 8.2 and were digested with LysC (FUJIFILM, 2 µl/100 µg protein) at RT for 3 h followed by tryptic digestion (0.5 µg/100 µg protein) at 37 °C overnight. Peptide mixtures were acidified with 1% trifluoroacetic acid and concentrated by vacuum centrifugation. Concentrates were further acidified by the addition of 100% acetic acid to a pH <2 and subsequently fractioned by C18-SCX custom-made stage tips as described before[39]. In short, samples were first loaded on pre-conditioned stage tips (2× SCX and 2× C18 disks) and eluted stepwise by increasing the NH4AcO concentration in elution buffers (20 mM to 500 mM NH4AcO in 0.5 % acetic acid and 20% acetonitrile). Collected fractions were desalted on custom-made C18 stage tips and solved in 0.1% formic acid.

## MS data collection and analysis

All samples were reconstituted in 0.1% formic acid and separated using an Easy-nLC1200 liquid chromatograph (Thermo Scientific) followed by peptide detection on a Q Exactive HF mass spectrometer (Thermo Scientific). Samples were separated on a 75 µm × 15 cm custom-made fused silica capillary packed with C18AQ resin (Reprosil-PUR 120, 1.9 µm, Dr. Maisch), flow rates and gradients were adjusted according to the experiment. Except for plasma membranome and whole proteome analysis, peptide mixtures were separated on a 35 min acetonitrile gradient in 0.1% formic acid at a flow rate of 400 nl/min (5–38% ACN gradient for 23 min, 38–60% ACN gradient for 3 min, 60–95% ACN gradient for 2 min). For plasma membranome analysis, peptide mixtures were separated on a 75 min acetonitrile gradient in 0.1% formic acid at a flow rate of 400 nl/min (7–38% ACN gradient for 53 min, 38–60% ACN gradient for 5 min, 60–100% ACN gradient for 5 min). For whole cell proteomics, peptide mixtures were separated on a 140 min acetonitrile gradient in 0.1% formic acid at a flow rate of 400 nl/min (3–6 % ACN gradient for 2 min, 6–30% ACN gradient for 90 min, 30–44% ACN gradient for 20 min, 44–75% ACN gradient for 10 min, 75–100% ACN gradient for 5 min). Peptides were ionized using a Nanospray Flex Ion Source (Thermo Scientific). Peptides were identified in full MS / dd MS² (Top15) mode, dynamic exclusion was enabled for 20 s and identifications with an unassigned charge or charges of one or >8 were rejected. For most analyses, MS1 resolution was set to 60,000 with a scan range of 300–1650 m/z, MS2 resolution to 15,000. For whole proteome analysis, MS1 resolution was set to 120,000 with a scan range of 300–1700 m/z, MS2 to 15,000. For all analyses, the AGC target1 was set to 3e6, AGC target2 to 1e5 and data collection controlled by Tune/Xcalibur (Thermo Scientific). Raw data were analyzed using MaxQuant's (version 1.6.0.1)[40] Andromeda search engine in reversed decoy mode based on a human reference proteome (Uniprot-FASTA, UP000005640, downloaded September 2017) with an FDR of 0.01 at both peptide and protein levels. Digestion parameters were set to specific digestion with trypsin with a maximum number of 2 missed cleavage sites and a minimum peptide length of 7. Oxidation of methionine and amino-terminal acetylation were set as variable and carbamidomethylation of cysteine as fixed modifications. The tolerance window was set to 20 ppm (first search) and to 4.5 ppm (main search). Depending on the experiment, label-free quantification (with a

minimum ratio count set to 2), re-quantification and match-between-runs was selected and at least 4 biological replicates were analyzed. Resulting protein group files were processed using Perseus (version 1.6.5.0)[41]. In general, common contaminants, reverse and site-specific identifications as well as proteins identified with a peptide count <1 or <2 were excluded. Remaining proteins were filtered by statistical testing as indicated. DAVID (version 6.8) was used for functional enrichment analysis. BP, biological process; CC, cellular component; MF, molecular function; I-Pro, inter pro; (UP) KW, uniprot key words; UP SEQ, uniprot sequence feature.

## ER-phagy flux assay

To measure the ER-phagy, 2000 293T cells stably expressing ssRFP-GFP-KDEL or mKeima-RAMP4 reporter per well were seeded in 384-well plates and incubated for 24 h before treatment. For cells expressing pCW57-CMV-ssRFP-GFP-KDEL, 1 µg/ml of Doxycycline (Sigma-Aldrich, D9891-1G) was added at the time of seeding cells to induce the expression of the reporter. 24 h post seeding, 50 µl of media containing either 0.1% DMSO or 250 nM Torin1 were added for indicated treatments. Screens in all three channels (fluorescence of GFP and RFP as well as cell confluence (phase)) were monitored every two hours for 60 h in total using an IncuCyte S3 (Sartorius). ER-phagy flux was calculated through changes in the ratio of the total fluorescence intensity of RFP/GFP. Line graphs represent the averaged ratio of data obtained from two biological replicates, each obtained from three individual wells (three technical replicates).

## Membrane fractionation

Cells from 10 × 15 cm dishes were harvested in ice-cold PBS and homogenized in 3 ml homogenization buffer (20 mM HEPES-KOH pH 7.2, 400 mM sucrose (Sigma), 1 mM EDTA, supplemented with protease inhibitors) with a douncer and subsequently subjected to differential centrifugation (protocol adapted from Ge et al.[42]). All steps were performed at 4 °C or on ice. Briefly, homogenates were first centrifuged at 1000 × g for 10 min to separate nuclei, followed by a centrifugation at 100,000 g for 30 min to pellet the total membrane. This was reconstituted in 0.75 mL 1.25 M sucrose buffer, overlayed with 0.5 mL 1.1 M and 0.5 mL 0.25 M sucrose buffer (all equilibrated with 20 mM Tricine/Tris pH 7.4 and supplemented with protease inhibitors). Sucrose gradients were centrifuged at 120,000 × g for 2 h and a light fraction (layer between the 0.25 mM and 1.1 mM fraction) and a pellet fraction (pellet at the tube bottom) were harvested. The light fraction was reconstituted in a 19% OptiPrep (Thermo) buffer and applied onto a 22.5 to 0% OptiPrep step-gradient. All OptiPrep densities were generated by mixing a 50% OptiPrep stock (OptiPrep in 20 mM Tricine pH 7.4, 42 mM sucrose, 1 mM EDTA) with a sucrose stock solution (250 mM sucrose, 20 mM Tricine pH 7.4, 1 mM EDTA). OptiPrep gradients were centrifuged at 150,000 × g for 3 h and six fractions collected from the top. All fractions were diluted with B88 buffer (20 mM HEPES-KOH pH 7.2, 250 mM sorbitol, 150 mM potassium acetate, 5 mM magnesium acetate, 0.3 mM DTT, 1× protease inhibitors and 1× PhosStop) and membranes pelleted by centrifugation at 100,000 g for 1 h. High-speed centrifugation steps were performed on an Optima™ Max-XP Tabletop Ultracentrifuge (Beckman Coulter) and an Optima L Ultracentrifuge (Beckman Coulter).

## Quantification and reproducibility

All presented micrographs and immunofluorescence images are representative for at least three individual experiments, showing similar results.

Immunofluorescence quantification and statistical analyses of at least 3 different experiments were performed using Columbus (PerkinElmer) or ImageJ Fiji, the exact number is stated in the respective figure legend. Data represents mean ± SEM, statistical significance was calculated by unpaired Student's t test. p-values <0.05 were considered

significant (<0.05 = *, <0.01 = ** and <0.001 = ***). Images were taken on an OPERA high-content screening platform (PerkinElmer) in 96 well-plates or on a Zeiss LSM800 with mounted cells. For analyses performed in Columbus (PerkinElmer), cells were segmented using "Find Nuclei B" and "Find Cytoplasm D" and filtered based on their cell area. Quantification of ERES followed with the identification of 488-spots using "Find Spots B" followed by selecting ERES spots via filtering for spot area, spot contrast, corrected spot intensity and relative spot intensity. Spots were normalized to cell number. For the analysis of ERES overlap, ERES spots (marker 1 in 488, marker 2 in 555) were selected via "Find Spots B". Both spot populations were merged individually, before an overlap region was defined using "Select region: Restrict by Mask". Ratios were calculated for overlap region areas compared to individual population areas. Trafficking of RUSH(-MAN2A1)-APEX2 was quantified by finding respective spots with "Find Spots B", which were subdivided into Golgi- and ER-like populations via filtering for spot area, spot contrast, corrected spot intensity, spot to region intensity and uncorrected spot intensity. Both populations were used for calculating the increase in Golgi-like signal distribution. For the analysis of RUSH cargos, spots were defined using "Find Spots B", followed by filtering for spot area, spot contrast, corrected spot intensity and spot to region intensity. Selected spots were either normalized to the cell number or, for GOLIM4 analysis, used to define the Golgi-spot free area that was further used to calculate the non-Golgi GOLIM4 intensity. The analysis of TMEM192 colocalization with lysosomal markers was performed in ImageJ Fiji. 488- and 555-signal areas were defined by automatically thresholding with the algorithm "Otsu". Manders' Coefficients were calculated based on channel intensities, Area overlap by defining regions interest using the "ROI manager".

### Reporting summary

Further information on research design is available in the Nature Portfolio Reporting Summary linked to this article.

## Data availability

All data are available upon request. All mass spectrometry / proteomic datasets reported in this study have been deposited to the ProteomeXchange Consortium via the PRIDE repository and are publicly available via the accession number PXD031874. A human reference proteome (Uniprot-FASTA, UP000005640, downloaded September 2017) was used for protein identification. Source data are provided with this publication.

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

## Acknowledgements

We thank Susanne Zellner for collaborative establishment of MS and EM protocols, members of the Behrends and Lichtenthaler labs as well as Hesso Farhan for critical discussions and shared reagents. We thank Michael Lazarou, Noboru Mizushima and Ivan Dikic for sharing reagents as well as Andrea Gubas and Sara Cano for technical support. This work was supported by the Deutsche Forschungsgemeinschaft (DFG, German Research Foundation) within the frameworks of the Munich Cluster for Systems Neurology (EXC 2145 SyNergy – ID 390857198 (C.B.), the Collaborative Research Center SFB 1177 (ID 259130777 (C.B., A.S.) and the project grant BE 4685/7-1 (C.B.) as well as by the Luke Heller TECPR2 Foundation (C.B.), the H2020-MSCA-ITN SAND (ID 860035) (C.B.), the Dr. Rolf M. Schwiete Stiftung (13/2017) (A.S.) and the Goethe University of Frankfurt (ENABLE) (A.S.). Z.E. was funded by the Legacy Heritage Fund (Grant #1935/16), the Israel Science Foundation (Grant #215/19), the Joint NSFC-ISF Research Fund (Grant #3345/20), and the Yeda-Sela Center for Basic Research.

## Author contributions

K.N. performed all experiments except EM preparation, which was conducted by M.S. and ER-phagy assays, which were performed by H.H. and A.S.; D.B., W.C.T. and L.A.W. generated HSAN9 cell models. S.L. contributed protocols and reagents. Z.E. provided critical discussions and edited the manuscript. All authors participated in data analysis and discussion. K.N. and C.B. conceived the study and wrote the manuscript.

## Funding

## Competing interests

The authors declare no competing interests.
