## [Peer Review File · Nature Communications]

REVIEWER COMMENTS

Reviewer #1 (Remarks to the Author):

In the submitted manuscript, the authors nicely combined RUSH and APEX labeling techniques to reveal the proteomic landscape change of vesicle transport machinery under the TECPR2 deficiency. Since the authors clearly presented their methods and results and the authors also carefully characterized their findings in systematic ways, I would like to recommend publishing this paper in Nature Communications after revision of a few minor things.

1. As shown in Figure 2a, the authors newly designed the RUSH-APEX2 system for selective mapping of ER lumen, COP vesicle lumen and golgi lumen proteome at the different pre-incubation conditions of biotin-phenol (BP). However, in the original article of APEX (Rhee et al. Science, 2013, 339, 1328-1331), pre-incubation of biotin-phenol at 37°C for 30 min was required for the evenly distribution of biotin-phenol throughout the intracellular membranes because the membrane permeability of biotin-phenol is not that great (discussed in Kang et al. Acc. Chem. Res. 2022, 55, 10, 1411–1422). Thus, pre-incubation of BP at 4°C for 30min or for 5 min at 37°C can affect the labeling efficiency of APEX2 because intracellular distribution of biotin-phenol could be perturbed at those conditions and Supplementary Fig. 3b (Streptavidin-HRP western blot result) also showed decreased BP labeling intensities at those conditions. Based on this WB result, I expected that the APEX protein coverage (e.g. total protein findings) can be also affected in each conditions but I'm still puzzled the APEX-labeled protein number at 5 min BP incubation showed higher number of proteins than that of 30 min of BP incubation (Fig. 2g). Thus, it would be great if the authors can provide their clear explanation and discussion on these results.

2. For the clearing cytosolic fractions, the authors utilized proteinase K treatment after lysis in this work. However, if the authors can find the BP-labeled tyrosine modification (Y+361 Da) in the APEX mass dataset, BP-modification can directly tell which proteins were modified by APEX and those new PTM information can also tell their luminal side localization if those proteins are transmembrane proteins. Thus, I'm wondering whether the authors can find some of this BP-modification information in their dataset (see Lee SY et al. J. Am. Chem. Soc. 2017,139, 3651-3662; Kang et al. Acc. Chem. Res. 2022, 55, 10, 1411–1422 for detailed information). It is noteworthy that some of BP-modification sites were also reported in the original workflow of APEX (Rhee et al. Science, 2013, 339, 1328-1331). Thus, it would be great if the authors can provide the additional BP-modification information in the revised materials.

Reviewer #2 (Remarks to the Author):

To understand TECPR2's role in protein cargo sorting and reveal consequences of disease mutation TECPR2 L440Rfs, the authors performed extensive proteomics characterizing various sites along the secretory pathway. TECPR2 had been previously reported to interact with ATG8s, modulate autophagy, interact with HOPS and BLOC, and stabilize COPII components. Here the authors have ample evidence to support TECPR2's function in ERES stabilization. However, the authors make two other conclusions that are not well supported by the data. First, they conclude (line 337) that "TECPR2's function within this pathway goes far beyond the initially proposed stabilization of COPII components." They claim TECPR2 also regulates ER to Golgi trafficking. The data supporting this claim (Figure 5) is not clear which I further detail in the points below. In the bulk of the paper, the authors measure protein levels at events that are downstream of ERES formation which can only further support TECPR2's role in ERES formation and gives little insight into other potential roles for TECPR2 in the secretory pathway. Second, the data gives little insight into TECPR2's link to autophagy. The authors fail to parse apart if the TECPR2 defect is solely from ERES destabilization and downstream mis-sorting of autophagy cargo or from a more direct interaction with autophagy machinery. To address this, the authors would need to have a more defined TECPR2 mutant that could not work in autophagy directly but preserves the stability of ERES.

I agree that the authors add a plentiful data resource for the TECPR2 field. They have identified protein cargoes whose sorting relies on full length TECPR2, which as the authors state in their discussion, can be potentially used for further mechanistic studies. In this study, a more complete characterization of the mis-trafficked proteins would give concrete guidance to the field as to which of these proteins should be further followed up on. In short, the mass spectrometry in this study is top of the line, but the subsequent cell biology including candidate follow-up and functional relevance is lacking experimental rigor. Here are each of my detailed concerns for the authors to address.

Concerns:

The results rely heavily on the use of one CRISPR clone of the TECPR2 L440Rfs truncation. Can a subset of the experiment be recapitulated by another clone, or can a subset of the experiments be rescued via reintroduction of the wildtype copy of TECPR2? Moreover, does another clone destabilize ERES and cause mis-sorting of the same cargoes/ or can destabilization of ERES or certain cargo sorting defects be rescued.

The model in figure 1b is misleading. The authors show COPII coat structures leaving the ER and arriving at the Golgi. Several sources of evidence in the field indicate that COPII never leaves the ER (Stephens et al JCS 2000, Shomron et al 2019 Biorxiv, Westrate et al JCS 2020, Weigel et al Cell 2021)

I appreciate that Figure 1 was a tremendous amount of proteomics work. Further verification of the proteomics would make this figure more complete. The microscopy colocalization analysis (Fig1g) lacks quantification. Also, it is difficult to detect/see the Sec31 staining. The authors should add image panels that are grayscale of each sec component on its own to understand the localization. Additionally, the authors do not have an example of an ERES component that they found to be enriched (USO1, GOLGA2, STX5 or RAB1B). It would also be beneficial to see localization of these potentially new ERES regulators (SPG20 and NEK9), but at least these identified protein candidates are confirmed by the authors via APEX-sec13/16 immunoblotting.

Can the authors clarify why they think SPG20 and NEK9 are other potential regulators of transport versus being client cargoes?

The heading “HSAN9-truncated TECPR2 affects ER-Golgi trafficking of COPII cargo” is not supported by their data. The data points to an ERES site destabilization (large fold changes RUSH-APEX comparing WT vs mutant at only 5min of release), but there is no evidence for TECPR2 directly regulating ER to Golgi trafficking. ERES destabilization results in delayed trafficking of cargo to the Golgi downstream, but based on your results, the mutant is more likely “affecting” the ER packaging and export step, not the trafficking from the ER to the Golgi.

To add something new to the field, the authors are identifying cargoes with delayed trafficking/potential mis-localization. This is largely based on one cargo-APEX (MAN2A1). The RUSH system has been reported to work differently with various RUSH cargoes (Boncompain et al 2014 protocol). Do the client proteins change when you use a different RUSH cargo? How do these cargoes behave similarly/differently?

Again, since this is a resource to the field identifying cargoes with delayed trafficking/potential mis-localization (figs 2-4), can the authors identify a class of cargoes that rely on full length TECPR2. Are there similarities and differences in the proteins that change? Why do some proteins still traffic as in the wildtype condition? What does this say about the function of TECPR2 and why ERES destabilization is detrimental to this potential set of proteins?

The “mis-localization” of candidates in Figure 2j are not quantified. Also, it is difficult to decipher how each protein localization is changing in the mutant condition based on these images.

Line 203. Is “membranome” the correct word here? The authors perform plasma membrane and lysosome membrane proteomics, but this doesn’t encompass the entire “membranome”.

The authors perform proteomics on secreted, plasma membrane, and lysosome proteins in wildtype vs TECPR2 L440Rfs to further emphasize that there are resulting downstream trafficking defects in the mutant. The plasma membrane and secreted protein measurements are sound and logical. However, for the Lyso-IP data, it is hard to conceptualize what a TMEM192 pull down means in the context of the TECPR2 L440Rfs. Based on HA IF microscopy images in Figure 4g the TMEM192 tag itself seems to be mis-trafficked. The authors need to further emphasize the potential flaw in not pulling down on a stable lysosome population in the mutant condition and discuss how the nature of the IP itself changes.

The authors should test if the slight overexpression of their TCEPR constructions affects localization of other trafficking components.

In figure 5e, it is difficult to see the level of each fluorescent protein in the zoom-in panels. Again, the authors need show images of each fluorescent channel.

In figure 5d, the APEX construct localizes to vesicular structures. The authors speculate at the identity of these membrane structures, but with this method the authors cannot tell if these are COPII structures potentially still continuous with the ER, or ERGIC vesicles, or Golgi vesicles.

Although TECPR2 full length (and not the mutant) associates with ER to Golgi, Golgi, and autophagy components, a potential mechanism for TECPR2 with respect to these proteins remains unclear. Using only this large truncation of TECPR2 makes defining a role in TECPR2 challenging. Potential mutations in certain regions of TECPR2 that do not result in this large ERES destabilization would aid in clarifying the role of TECPR2. For instance, how does a ATG8 interacting mutant affect trafficking of other proteins and ERES stabilization? In addition, inhibiting ATG8 lipidation may not destabilize the TECPR2-ATG8 interaction which could result in TECPR2 still being on the membrane. The authors did not measure if inhibiting ATG8 lipidation resulted in trafficking defects like those seen for the TECPR2 mutant.

Reviewer #3 (Remarks to the Author):

This article used the finest modern proteomics approach to show how a pathological mutation in TECPR2 could perturb the secretory pathway. They used proximity biotinylation to demonstrate that mutant TECPR2 affects the assembly of ERES components, proximity biotinylation coupled to the RUSH assay (very smart assay!) to show it affects ER to Golgi trafficking of COPII cargoes, click-chemistry to analyse secreted glycoproteins and showed alteration in the secretome and surfaceome, lysolIP and proteomics to show defects in lysosome content and IP to show that it affects partners of TECPR2. Other approaches such as western blotting systematically verified the most relevant hits.

The experiments were rigorously carried out, the results are clearly presented and fairly discussed. This is a solid piece of work and the systematic approach presented here is certainly a blueprint for further investigations of secretory defects in rare diseases.

This reviewer has only one comment/suggestion. Indeed, the authors assume that the secretome results only (mainly?) from the conventional secretory pathway (ER-Golgi-Surface). In fact, the defects they found, including in lysosomes, surely also affect unconventional secretion such as that mediated by late endosomes. In addition, glycosylation mainly occurs in the ER and released glycosylated proteins could have bypassed the Golgi apparatus. When looking at the spreadsheet, this reviewer could identify some proteins known to be released by unconventional secretion. Thus, the authors should add further analysis of their secretomics data to clearly look at unconventional secretion. With these additional data, this reviewer would consider this article as a landmark in the field.

We would like to thank the reviewers for their time and greatly appreciate their constructive and overall positive feedback. We are confident that the changes we made substantially improved our paper and qualify our findings for publication.

REVIEWER #1:

In the submitted manuscript, the authors nicely combined RUSH and APEX labeling techniques to reveal the proteomic landscape change of vesicle transport machinery under the TECPR2 deficiency. Since the authors clearly presented their methods and results and the authors also carefully characterized their findings in systematic ways, I would like to recommend publishing this paper in Nature Communications after revision of a few minor things.

1. As shown in Figure 2a, the authors newly designed the RUSH-APEX2 system for selective mapping of ER lumen, COP vesicle lumen and golgi lumen proteome at the different pre-incubation conditions of biotin-phenol (BP). However, in the original article of APEX (Rhee et al. Science, 2013, 339, 1328-1331), pre-incubation of biotin-phenol at 37°C for 30 min was required for the evenly distribution of biotin-phenol throughout the intracellular membranes because the membrane permeability of biotin-phenol is not that great (discussed in Kang et al. Acc. Chem. Res. 2022 , 55, 10, 1411–1422). Thus, pre-incubation of BP at 4°C for 30min or for 5 min at 37°C can affect the labeling efficiency of APEX2 because intracellular distribution of biotin-phenol could be perturbed at those conditions and Supplementary Fig. 3b (Streptavidin-HRP western blot result) also showed decreased BP labeling intensities at those conditions. Based on this WB result, I expected that the APEX protein coverage (e.g. total protein findings) can be also affected in each conditions but I'm still puzzled the APEX-labeled protein number at 5 min BP incubation showed higher number of proteins than that of 30 min of BP incubation (Fig. 2g). Thus, it would be great if the authors can provide their clear explanation and discussion on these results.

We agree that based on the poor membrane permeability of BP, most laboratories perform a 30 min incubation step to ensure even distribution and efficient labeling. In the mentioned publication by Rhee et al. 2013, the Ting lab used APEX to perform proteomic mapping of mitochondria. In our study, we used the improved peroxidase APEX2 (first described in Lam et al. Nat Methods 12, 51-54 (2015)) that showed a higher biotinylation activity than APEX, enabling a more efficient labelling even at potentially lower BP saturations. Secondly, based on the different nature of the double membrane of mitochondria compared to the ER membrane, it is likely that the ER lumen can be more readily saturated. However, we agree that the 5 min incubation conditions should result in a lower amount of biotinylated proteins than the 30 min labeling conditions. The concern raised regarding this issue was based on renumbered Fig. 3g (previous Fig. 2g) which is not displaying the number of enriched proteins over background but the proteins changed between TECPR2 WT and MUT cells in the same labelling condition. Hence, the higher number of proteins displayed in renumbered Fig. 3g (previous Fig. 2g) is more suited to compare the effect of TECPR2 variants (where 5 min trafficking had the most striking differences) and not the changes between different timepoints. Comparing the total amount of enriched proteins per labeling condition in TECPR2 WT cells compared to 0 min BP controls shows a clear increase in protein numbers (represented in renumbered Suppl. Fig. 4e,h (previous Suppl. Fig. 3e,h)). As expected, 30 min of BP incubation resulted in in the strongest protein enrichment, followed by 5 min at 37 °C and finally 30 min at 4 °C (where the enzymatic activity is decreased due to low temperatures).

2. For the clearing cytosolic fractions, the authors utilized proteinase K treatment after lysis in this work. However, if the authors can find the BP-labeled tyrosine modification (Y+361 Da) in the APEX mass dataset, BP-modification can directly tell which proteins were modified by APEX and those new PTM information can also tell their luminal side localization if those proteins are transmembrane proteins. Thus, I'm wondering whether the authors can find some of this BP-modification information in their dataset (see Lee SY et al. J. Am. Chem. Soc. 2017,139, 3651-3662; Kang et al. Acc. Chem. Res. 2022 , 55, 10, 1411–1422 for detailed information). It is noteworthy that some of BP-modification sites were also reported in the original workflow of APEX (Rhee et al. Science, 2013, 339, 1328-1331). Thus, it would be great if the authors can provide the additional BP-modification information in the revised materials.

We thank the reviewer for this really interesting idea. Based on the suggested publications, we reanalyzed our RUSH-APEX dataset with the respective tyrosine modifications (Y+361 and Y+377). Unfortunately, we did not detect a substantial number of BP-modified sites. One possible explanation could be that in Lee et al. 2017, peptides were eluted by boiling the beads in formamide. In our method, we performed an on-bead tryptic digest. This leaves the possibility that the actual BP-modified site is still bound to the beads. To test this, we performed a pilot experiment using our RUSH (MAN2A1)-APEX2 TECPR2 WT cells and processed samples following our standard protocol i) with no additional elution, ii) with an additional elution using formamide or iii) with an additional elution in 80 % ACN supplemented with TFA and FA. Results can be seen below.

Figure 1 Analysis of biotinylated peptides. (a) Indicates the different conditions. (b) Quality control values for identified protein groups, peptides and intensities. (c) Identified Y+361 sites and protein groups. (d) GO-Term analysis for these protein groups.

In sum, we did not detect many BP-modified peptides/protein groups (13 with Y+361, non for Y+377, in total 19 sites), indicating a general incompatibility of our sample procession method. However, the identified BP-modified proteins were all present in our original dataset, found to be affected by TECPR2 deficiency and comprised early secretory pathway compartment-specific GO terms. Hence, this supports our view that combining APEX2 targeting to the ER lumen with proteinase K treatment allows to specifically enrich proteins found in the early secretory pathway. Nevertheless, given the low number of identifications we decided not to include this analysis in our current manuscript.

REVIEWER#2:

To understand TECPR2's role in protein cargo sorting and reveal consequences of disease mutation TECPR2 L440Rfs, the authors performed extensive proteomics characterizing various sites along the secretory pathway. TECPR2 had been previously reported to interact with ATG8s, modulate autophagy, interact with HOPS

and BLOC, and stabilize COPII components. Here the authors have ample evidence to support TECPR2's function in ERES stabilization. However, the authors make two other conclusions that are not well supported by the data. First, they conclude (line 337) that "TECPR2's function within this pathway goes far beyond the initially proposed stabilization of COPII components." They claim TECPR2 also regulates ER to Golgi trafficking. The data supporting this claim (Figure 5) is not clear which I further detail in the points below. In the bulk of the paper, the authors measure protein levels at events that are downstream of ERES formation which can only further support TECPR2's role in ERES formation and gives little insight into other potential roles for TECPR2 in the secretory pathway.

We agree that our data mainly pinpoint towards defects at ERES or the ER-side of the ER-Golgi interface and thus toned down the aforementioned statement in the current version of our manuscript. However, we would like to mention that our proteomics and biochemical experiments (Fig. 6, Fig. 7, Suppl. Fig. 8) point to an extended TECPR2 interaction landscape within the early secretory pathway. A detailed response to individual concerns can be found below.

Second, the data gives little insight into TECPR2's link to autophagy. The authors fail to parse apart if the TECPR2 defect is solely from ERES destabilization and downstream mis-sorting of autophagy cargo or from a more direct interaction with autophagy machinery. To address this, the authors would need to have a more defined TECPR2 mutant that could not work in autophagy directly but preserves the stability of ERES.

We agree that our manuscript does not add substantial new data on TECPR2's role in autophagy. However, our primary goal was to systematically assess trafficking defects with a special focus on ER export as this has not been examined yet. Nevertheless, understanding the exact relationship between defects in ERES and early autophagosome formation in TECPR2 deficiency is an interesting topic. Towards this longterm goal, we re-examined the role of human ATG8 proteins in the context of TECPR2 and found that they were dispensable for its association with cellular membranes. In addition, we examined to what extent TECPR2 deficiency induces ER-phagy and failed to detect an induction of this pathway under basal (fed) growth conditions using two different reporter systems. This additional data is provided as new Fig. 7c and new Suppl. Fig. 7h. Moreover, we are currently mapping the binding regions of different TECPR2 interaction partners to test whether autophagy- and ER export-related functions of TECPR2 can be taken apart. However, since this requires extensive additional efforts, we find this to be out of the scope of the preset study and its worth a more detailed analysis.

I agree that the authors add a plentiful data resource for the TECPR2 field. They have identified protein cargoes whose sorting relies on full length TECPR2, which as the authors state in their discussion, can be potentially used for further mechanistic studies. In this study, a more complete characterization of the mis-trafficked proteins would give concrete guidance to the field as to which of these proteins should be further followed up on. In short, the mass spectrometry in this study is top of the line, but the subsequent cell biology including candidate follow-up and functional relevance is lacking experimental rigor. Here are each of my detailed concerns for the authors to address.

Concerns:

The results rely heavily on the use of one CRISPR clone of the TECPR2 L440Rfs truncation. Can a subset of the experiment be recapitulated by another clone, or can a subset of the experiments be rescued via reintroduction of the wildtype copy of TECPR2? Moreover, does another clone destabilize ERES and cause mis-sorting of the same cargoes/ or can destabilization of ERES or certain cargo sorting defects be rescued.

We thank the reviewer for raising this important point. To confirm that some of the TECPR2-dependent disturbances can also be observed in other cell lines or clones, we performed additional experiments addressing this concern: Regarding the disintegration of ERES, we were able to rescue the reduction of ERES by the overexpression of full-length but not disease-mutant TECPR2 (new Fig. 2a,b) and did observe a similar effect on ERES in patient-derived fibroblasts (new Fig. 2c,d). Concerning the SUSPECS and SPECS data, we added a new candidate validation in patient-derived fibroblasts (new Suppl. Fig. 6a and 6f). Additionally, we would like to emphasize that both proteomic datasets were acquired using cell lines expressing endogenous TECPR2 WT and MUT as well as in TECPR2 MUT cells reconstituted with HA-tagged WT or L440Rfs TECPR2. We then performed a cross analysis and only selected the candidates that are similarly altered comparing TECPR2 WT vs MUT and HA-TECPR2 WT vs HA-TECPR2 L440Rfs (renumbered Fig 4b).

The model in figure 1b is misleading. The authors show COPII coat structures leaving the ER and arriving at the Golgi. Several sources of evidence in the field indicate that COPII never leaves the ER (Stephens et al JCS 2000, Shomron et al 2019 Biorxiv, Weststrate et al JCS 2020, Weigel et al Cell 2021).

We fully agree and have adjusted all schematics accordingly.

I appreciate that Figure 1 was a tremendous amount of proteomics work. Further verification of the proteomics would make this figure more complete. The microscopy colocalization analysis (Fig1g) lacks quantification. Also, it is difficult to detect/see the Sec31 staining. The authors should add image panels that are grayscale of each sec component on its own to understand the localization.

As suggested, we added both quantification and individual images verifying the differential decrease per ERES component. This is now shown in the revised Fig. 2e (previous Fig. 1g). The quantification was performed by measuring the relative overlap of both signal areas.

Additionally, the authors do not have an example of an ERES component that they found to be enriched (USO1, GOLGA2, STX5 or RAB1B). It would also be beneficial to see localization of these potentially new ERES regulators (SPG20 and NEK9), but at least these identified protein candidates are confirmed by the authors via APEX-sec13/16 immunoblotting. Can the authors clarify why they think SPG20 and NEK9 are other potential regulators of transport versus being client cargoes?

To address this concerns, we performed additional biochemical validation of these cis-Golgi proteins in TECPR2 WT and MUT cells expressing APEX2-SEC13 (new Fig. 2f). These pulldown experiments confirmed the increased SEC13 proximity of GOLGA2, STX5 and USO1. Moreover, we performed protease protection assay to address whether NEK9, SPG20 or BAG2 are client cargoes. However, this analysis unveiled that all three proteins were not protected from

proteinase K digest and thus most likely present in the cytosol (new Fig. 2i), indicating that the increased proximity of COPII components and BAG2, NEK9 and SPG20 occurs in the cytoplasm. This is consistent with the lack of signal peptide sequences in these proteins. For the colocalization analysis of COPII proteins with NEK9 and SPG20, we had to rely on transient overexpression of these candidates due to a lack of suitable antibodies. The respective colocalization experiments (an example is shown below for HA-tagged NEK9 and endogenously stained SEC13 and SEC16A) failed to show a clear colocalization due to widespread signal of both candidates throughout the cytosol and the reduced presence of distinct COPII structures in TECPR2 deficient cells.

Given the lack of a distinct subcellular colocalization of NEK9 and SPG20 upon TECPR2 deficiency, we restrain from classifying both proteins as potential new ERES regulators and adjusted the manuscript accordingly. More likely, the increased proximity of both proteins could be a result of the disintegration of ERES. In this scenario, COPII coat proteins would re-distribute to the cytosol rather than NEK9 and SPG20 getting recruited to ERES.

The heading “HSAN9-truncated TECPR2 affects ER-Golgi trafficking of COPII cargo” is not supported by their data. The data points to an ERES site destabilization (large fold changes RUSH-APEX comparing WT vs mutant at only 5min of release), but there is no evidence for TECPR2 directly regulating ER to Golgi trafficking. ERES destabilization results in delayed trafficking of cargo to the Golgi downstream, but based on your results, the mutant is more likely “affecting” the ER packaging and export step, not the trafficking from the ER to the Golgi.

As mentioned above, we agree that our data mainly point to defects at ERES or the ER-side of the ER-Golgi interface. We adjusted the heading and all statements accordingly.

To add something new to the field, the authors are identifying cargoes with delayed trafficking/potential mis-localization. This is largely based on one cargo-APEX (MAN2A1). The RUSH system has been reported to work differently with various RUSH cargoes (Boncompain et al 2014 protocol). Do the client proteins change when you use a different RUSH cargo? How do these cargoes behave similarly/differently?

We thank the reviewer for this idea and agree that using additional RUSH-reporters to profile their cargo would clearly advance our understanding of the effects of TECPR2 deficiency on COPII cargo, especially with regard to potential subpopulations of cargo carriers or cargo types. Unfortunately, we did not manage to acquire a second cargo profiling dataset with a different RUSH reporter. We tried incorporating APEX2 in several RUSH plasmids carrying GPI, ssEGFP or TNFalpha. Cloning of an such an APEX2-RUSH fusion plasmid only

worked for TNFalpha. However, all attempts to generate stably expressing cell lines (similarly to MAN2A1) were not successful as cells regularly died or did not express the APEX2-TNFalpha-RUSH chimera. Therefore, we tried transient overexpression of APEX2-TNFalpha-RUSH. Although initial immunoblots showed promising results immunostaining revealed a very low transfection rate (see figure below, panel a and b). We still decided to proceed with a mass spectrometry analysis but faced severe problems with a high detection variance of TNFalpha itself as well as overall identified proteins. In general, identified proteins groups were dramatically low compared to the MAN2A1 dataset. By analysing a pilot experiment in which cells expressing APEX2-TNFalpha-RUSH for 24 h were incubated with BP for 30 min before performing biotinylation, we detected some of our top cargo candidates identified in the MAN2A1 dataset (e.g., CLPTM1L, ERGIC1, GOLIM4, LGALS3BP) and observed enrichment of similar secretory pathways-related GO terms (panel c and d). Due to uneven bait levels, LFQ intensities had to be adjusted before comparison of TECPR2 WT vs MUT cells. In doing so, no clear effect of TECPR2 deficiency was observable for CLPTM1L and ERGIC1 (panel c). LGALS3BP and GOLIM4 showed the same tendency though with a smaller effect size as in the MAN2A1 dataset. Given the described problems in establishing a consistent and comparable overexpression of APEX2-TNFalpha-RUSH, we concluded that this alternative RUSH-APEX2 reporter did not match our quality control criteria and therefore decided not to include the respective data in the current manuscript.

Again, since this is a resource to the field identifying cargoes with delayed trafficking/potential mis-localization (figs 2-4), can the authors identify a class of cargoes that rely on full length TECPR2. Are there similarities and differences in the proteins that change? Why do some proteins still traffic as in the wildtype condition? What does this say about the function of TECPR2 and why ERES destabilization is detrimental to this potential set of proteins?

We agree that narrowing-down protein cargo classes being affected by TECPR2 deficiency would increase impact and guidance for further research. Thereto, we focused on extracting unique features from the results of the COPII content

profiling as well as the SPECS and SUSPECS datasets since these approaches monitored primarily disturbances in the secretory pathway. When analyzing respective GO-terms, we found a rather wide range of protein classes to be affected. This points toward a more general trafficking and secretory disturbance and would comply with the observed general disintegration of ERES. However, there were some GO-terms consistently affected by TECPR2 deficiency across datasets, namely integrin binding, laminin binding, laminin, fibronectin, EGF-containing proteins and plexins/SEMA-containing proteins. All of those are important factors for the cell's extracellular matrix and interaction with its environment, some of them with annotated functions in the neuronal context. To show the enrichment of these structural elements across datasets, we updated the GO analysis of the COPII content profiling by including the respective SMART terms (revised Fig 3i) and expanded the SPECS/SUSPECS result section to highlight the occurrence of similar terms in these datasets (page 10, line 259 ff).

The “mis-localization” of candidates in Figure 2j are not quantified. Also, it is difficult to decipher how each protein localization is changing in the mutant condition based on these images.

We expanded the old Fig. 2j (now revised Fig 3j) by adding quantification for each analyzed protein. For ERGIC1, CLPTM1L and LGALS3BP, this was done by counting spots per cell in an automated unbiased manner; for GOLIM4 the summed intensity in the cytoplasm (non-Golgi area) was measured.

Line 203. Is “membranome” the correct word here? The authors perform plasma membrane and lysosome membrane proteomics, but this doesn't encompass the entire “membranome”.

We thank the reviewer for pointing this out and changed this term to “cell surface proteome” throughout the manuscript.

The authors perform proteomics on secreted, plasma membrane, and lysosome proteins in wildtype vs TECPR2 L440Rfs to further emphasize that there are resulting downstream trafficking defects in the mutant. The plasma membrane and secreted protein measurements are sound and logical. However, for the Lyso-IP data, it is hard to conceptualize what a TMEM192 pull down means in the context of the TECPR2 L440Rfs. Based on HA IF microscopy images in Figure 4g the TMEM192 tag itself seems to be mis-trafficked. The authors need to further emphasize the potential flaw in not pulling down on a stable lysosome population in the mutant condition and discuss how the nature of the IP itself changes.

Given the lysosomal phenotype described by the Elazar group (Fraiberg et al. Autophagy. 2021; 17(10): 3096–3108), this is a valid concern. Although there is no indication of leaky or overall instable lysosomes, we cannot exclude that we enrich a off-pathway compartment to which TMEM192 has been mis-trafficked . To exclude such a scenario, we performed quantification of the overlap of 3xHA-TMEM192 with endogenous LAMP1 and LAMP2 positive compartments, respectively. In doing so, we did not detect changes in Mander's coefficients or area overlaps comparing TECPR2 WT and MUT cells (revised Fig. 5b,c). Similarly, we also observed extensive colocalization of tagged TMEM192 with endogenous CD63 in TECPR2 WT and MUT cells (revised Fig. 5j). Thus, we

concluded that tagged TMEM192 is efficiently sorted to lysosomes in both conditions.

The authors should test if the slight overexpression of their TCEPR constructions affects localization of other trafficking components.

To address this point, we monitored the Golgi marker Giantin and the ER marker Calnexin in TECPR2 WT and MUT cells as well as in TECPR2 MUT cells re-expressing HA-TECPR2 WT or L440Rfs. Across this panel, we did not observe any overt differences. This additional data is provided as new Suppl. Fig. 3g.

In figure 5e, it is difficult to see the level of each fluorescent protein in the zoom-in panels. Again, the authors need show images of each fluorescent channel.

As requested, we expanded the previous Fig. 5e (now revised Fig. 6e) by individual fluorescent channels.

In figure 5d, the APEX construct localizes to vesicular structures. The authors speculate at the identity of these membrane structures, but with this method the authors cannot tell if these are COPII structures potentially still continuous with the ER, or ERGIC vesicles, or Golgi vesicles.

The reviewer is absolutely right. As it is quite difficult to distinguish distinct compartments within the ER to Golgi interface, we performed a multi-step differential centrifugation approach that is able to further differentiate at least the ER and ERES from the Golgi (new Fig. 7e,f). In doing so, we detected TECPR2 prominently in several membranous fractions enriched for ER, ERES and ERGIC markers whereas Golgi-enriched fractions showed significantly reduced TECPR2 signal (Fig. 7f). Based on these new results in combination with our interaction and proximity studies, we propose that TECPR2 is associated with components of the ER-Golgi interface, predominantly early ER-associated structures.

Although TECPR2 full length (and not the mutant) associates with ER to Golgi, Golgi, and autophagy components, a potential mechanism for TECPR2 with respect to these proteins remains unclear. Using only this large truncation of TECPR2 makes defining a role in TECPR2 challenging. Potential mutations in certain regions of TECPR2 that do not result in this large ERES destabilization would aid in clarifying the role of TECPR2. For instance, how does a ATG8 interacting mutant affect trafficking of other proteins and ERES stabilization? In addition, inhibiting ATG8 lipidation may not destabilize the TECPR2-ATG8 interaction which could result in TECPR2 still being on the membrane. The authors did not measure if inhibiting ATG8 lipidation resulted in trafficking defects like those seen for the TECPR2 mutant.

We thank the reviewer for this comment regarding differentiating the effects of TECPR2 deficiency with respect to its involvement in ATG8-binding and autophagy. However, we think that adequately addressing these open questions would require substantial additional experiments and would therefore be beyond the scope of this resource-like manuscript. Nevertheless, we confirmed the results of ATG7 inhibitor and found that TECPR2 is still associated with membranes in cells lacking all six human ATG8 proteins. This result is now provided as new Fig. 7d.

REVIEWER #3:

This article used the finest modern proteomics approach to show how a pathological mutation in TECPR2 could perturb the secretory pathway. They used proximity biotinylation to demonstrate that mutant TECPR2 affects the assembly of ERES components, proximity biotinylation coupled to the RUSH assay (very smart assay!) to show it affects ER to Golgi trafficking of COPII cargoes, click-chemistry to analyse secreted glycoproteins and showed alteration in the secretome and surfaceome, lysolP and proteomics to show defects in lysosome content and IP to show that it affects partners of TECPR2. Other approaches such as western blotting systematically verified the most relevant hits.

The experiments were rigorously carried out, the results are clearly presented and fairly discussed. This is a solid piece of work and the systematic approach presented here is certainly a blueprint for further investigations of secretory defects in rare diseases. This reviewer has only one comment/suggestion. Indeed, the authors assume that the secretome results only (mainly?) from the conventional secretory pathway (ER-Golgi-Surface). In fact, the defects they found, including in lysosomes, surely also affect unconventional secretion such as that mediated by late endosomes. In addition, glycosylation mainly occurs in the ER and released glycosylated proteins could have bypassed the Golgi apparatus. When looking at the spreadsheet, this reviewer could identify some proteins known to be released by unconventional secretion. Thus, the authors should add further analysis of their secretomics data to clearly look at unconventional secretion. With these additional data, this reviewer would consider this article as a landmark in the field.

We would like to thank the reviewer for their positive review. Due to the previously described interaction of TECPR2 with the COPII coat component SEC24D, we mainly focused on conventional COPII-mediated secretion. As SUSPECS/SPECS takes advantage of glycosylation in the ER (and to a lower extent the Golgi), we expected to predominantly detect conventionally secreted proteins. To highlight this aspect, we added a third GO term pie chart to the Suppl. Fig. 5i, displaying the percentage of signal peptide containing SPECS proteins. This sequence targets proteins to the ER and was present in over 80 % of SPECS hits. However, it is indeed plausible that TECPR2 deficiency might also affect unconventionally secreted proteins (UPS) – possibly as a result of disturbances in the endo-lysosomal system. In any case, we re-analyzed our SPECS dataset with a special focus on classical UPS targets as suggested by this reviewer. In an initial manual analysis of SPECS proteins with well-established UPS candidates (ANXA1, ANXA2, ARF6, CD63, CD81, CD9, CFTR, EGFR1, EN2, ENO1, ENO2, ENO3, FABP4, FGF1, FGF2, FGF9, FLOT1, FLOT2, GAPDH, GRASP55, GRASP65, GRP94, GRP96, HMGB1, HSP70, HSP90, IL1B, LGALS3, MAPT, MIF, PGK1, SLC26A4, SNCA, SOD1, TG2 and TSG101), only one candidate was present (HSP90). However, HSP90 was not similarly affected in cells expressing endogenous/exogenous TECPR2 variants. Moreover, we performed a cross comparison with two recent publications investigating the proteome of two different UPS pathways: (i) An ATG8-dependent extracellular vesicle proteome by the Debnath group (Leidal et al. Nat Cell Biol 22, 187–199 (2020)) and (ii) an exosome proteome by the Kalluri group (Kugeratski et al. Nat Cell Biol 23, 631–641 (2021)). In both comparisons, only a small proportion of our 313 SPECS proteins was present in each dataset and an even smaller number was affected by TECPR2. This additional analysis is provided as new Suppl. Fig. 6g and 6h). Taken together and given potential limitations of the SPECS approach, our data pinpoint to a prominent disturbance of conventionally

secreted proteins and only marginal effect on UPS. Nevertheless, the latter will be interesting to follow up in future studies.

REVIEWERS' COMMENTS

Reviewer #2 (Remarks to the Author):

I commend your efforts in dissecting TECPR2's role in ER export and support publication after a few minor adjustments. For the imaging analysis work, the authors need to add to the methods how the spots were identified and what segmentation strategy they used to identify regions in each image for the overlay analysis. I also recommend labelling the insets of Fig 5e expanding panels with WT or mutant to add clarity.

In short, my concerns were addressed, and I recommend publication. Here are my comments to each of the original points of my reviews with the rebuttal points:

I find the results in Figure 7c and sup fig 7h to give insight into my question about TECPR2's potential role in inducing ER-phagy mechanisms. I am excited to see your further work in this area and I agree that full characterization of interaction partners is outside the scope of this study.

The authors tested TECPR2 in other cell lines and performing rescue validation as I requested.

The authors correctly adjusted the schematics.

The image quantification adds to the thorough analysis. The authors need to add to the methods how the spots were identified and what segmentation strategy they used to identify regions in each image for the overlay analysis.

The authors sufficiently addressed my question and I agree with the reclassification of NEK9 and SPG20 and the model that COPII coat proteins redistribute to the cytosol.

The authors adjusted the heading to better fit their data.

I commend the authors efforts in this and realize the limitations of the RUSH system. I agree that testing of additional cargo does not take away from their study and findings and should not take away from publication of this work.

As requested, the authors investigated the protein cargo classes affected by TECPR2, and I agree with the authors that their data points towards a wide-range effect on ERES. The uncovering of consistent disruption of extracellular matrix components is an interesting addition that adds to the authors' work.

The authors successfully performed the analysis I suggested. Again, just add more information to the methods as to how spots were identified.

The authors adjusted the heading to better fit their data.

The authors confirmed their tag is at lysosomal structures for their lyso-IP analysis.

The authors confirmed Golgi and ER are intact.

The authors clearly showed their results by expanding the individual channels. As a minor change, the authors need to label the inset expanding panels with WT or mutant to add clarity.

I commend your efforts using differentiation centrifugation to address my question. The authors revealed TECPR2 enrichment at ER-Golgi and ER-associated structures.

The authors tested the requirement for ATG8 proteins and found TECPR2 is still recruited. I agree that this is sufficient for the scope of their manuscript.

We would like to thank Reviewer #2 for her/his positive evaluation and are glad that we were able to improve our manuscript to clear any previous concerns. Upon making the requested final modifications, we are now very much looking forward to publication and providing our findings to the scientific community.

Reviewer #2 (Remarks to the Author):

I commend your efforts in dissecting TECPR2's role in ER export and support publication after a few minor adjustments. For the imaging analysis work, the authors need to add to the methods how the spots were identified and what segmentation strategy they used to identify regions in each image for the overlay analysis.

We thank the reviewer for this note and agree that this will help to increase reproducibility. We therefore added additional information to the methods section.

I also recommend labelling the insets of Fig 5e expanding panels with WT or mutant to add clarity.

As recommended, we modified the labelling of Fig 5e insets.

In short, my concerns were addressed, and I recommend publication. Here are my comments to each of the original points of my reviews with the rebuttal points:

I find the results in Figure 7c and sup fig 7h to give insight into my question about TECPR2's potential role in inducing ER-phagy mechanisms. I am excited to see your further work in this area and I agree that full characterization of interaction partners is outside the scope of this study.

The authors tested TECPR2 in other cell lines and performing rescue validation as I requested.

The authors correctly adjusted the schematics.

The image quantification adds to the thorough analysis. The authors need to add to the methods how the spots were identified and what segmentation strategy they used to identify regions in each image for the overlay analysis.

As requested, details on image analysis were added to the methods section.

The authors sufficiently addressed my question and I agree with the reclassification of NEK9 and SPG20 and the model that COPII coat proteins redistribute to the cytosol.

The authors adjusted the heading to better fit their data.

I commend the authors efforts in this and realize the limitations of the RUSH system. I agree that testing of additional cargo does not take away from their study and findings and should not take away from publication of this work.

As requested, the authors investigated the protein cargo classes affected by TECPR2, and I agree with the authors that their data points towards a wide-range effect on ERES. The uncovering of consistent disruption of extracellular matrix components is an interesting addition that adds to the authors' work.

The authors successfully performed the analysis I suggested. Again, just add more information to the methods as to how spots were identified.

As requested, details on image analysis were added to the methods section.

The authors adjusted the heading to better fit their data.

The authors confirmed their tag is at lysosomal structures for their lyso-IP analysis.

The authors confirmed Golgi and ER are intact.

The authors clearly showed their results by expanding the individual channels. As a minor change, the authors need to label the inset expanding panels with WT or mutant to add clarity.

As recommended, we modified the labelling of Fig 5e insets.

I commend your efforts using differentiation centrifugation to address my question. The authors revealed TECPR2 enrichment at ER-Golgi and ER-associated structures.

The authors tested the requirement for ATG8 proteins and found TECPR2 is still recruited. I agree that this is sufficient for the scope of their manuscript.